

# Carbon Stocks and Accumulation Rates in Salt Marshes of the Pacific Coast of Canada

Stephen G. Chastain[1], Karen Kohfeld[1], Marlow G. Pellatt[1,2]

[1]Simon Fraser University School of Resource & Environmental Management, Burnaby, Canada V5A 1S6
5 [2]Parks Canada, Protected Areas Establishment and Conservation Directorate, Vancouver, British Columbia, Canada V6B 6B4

*Correspondence to*: Stephen G. Chastain (Stevegeo90@gmail.com)



**Abstract**

Tidal salt marshes are known to accumulate "blue carbon" at high rates relative to their surface area and have been put forth as a potential means for enhanced CO2 sequestration. However, estimates of salt marsh carbon accumulation rates are based on a limited number of marshes globally and the estimation of carbon accumulation rates require detailed dating to provide accurate estimates. We address one data gap along the Pacific Coast of Canada by estimating carbon stocks in 34 sediment cores and estimating carbon accumulation rates using $^{210}$Pb dating on four cores from seven salt marshes within the Clayoquot Sound UNESCO Biosphere Reserve and Pacific Rim National Park Reserve of Canada (49.2° N, 125.80° W). Carbon stocks averaged $80.6 \pm 43.8$ megagrams of carbon per hectare (Mg C ha$^{-1}$) between the seven salt marshes, and carbon accumulation rates averaged $146 \pm 102$ grams carbon per square meter per year (g C m$^{-2}$ yr$^{-1}$). These rates are comparable to those found in salt marshes further south along the Pacific coast of North America (32.5-38.2° N) and at similar latitudes in Eastern Canada and Northern Europe (43.6-55.5° N). The seven Clayoquot Sound salt marshes currently accumulate carbon at a rate of 54.28 Mg C yr$^{-1}$ over an area of 46.94 ha, 87 % of which occurs in the high marsh zone. On a per-hectare basis, Clayoquot Sound salt marsh soils accumulate carbon at least one order of magnitude more quickly than the average of global boreal forest soils, and approximately two times larger than rates for forests in British Columbia. However, because of their relatively small area, we suggest that their carbon accumulation rate capacity could best be considered as a climate mitigation co-benefit when conserving for other salt marsh ecosystem services.




## 1. Introduction

Coastal, vegetated ecosystems, such as eelgrass meadows, mangroves, and tidal salt marshes, have recently been recognized for their ability to store large amounts of carbon, or "blue carbon," within their soils and sediments (IPCC 2014; Howard et al. 2014). While blue carbon ecosystems cover approximately 0.2 % of the ocean surface, previous studies have
suggested that they could be responsible for up to 50 % of total ocean carbon burial (Duarte et al. 2005) and that their per-area carbon sequestration rate is substantially greater than that of terrestrial forests (McLeod et al. 2011). Globally, blue carbon ecosystems have been estimated to sequester between 75.3 and 224.2 Tg C $yr^{-1}$ (Duarte et al. 2013).

The high carbon storage and accumulation capacity per-area of coastal ecosystems have been investigated because of the potential for blue carbon to provide climate mitigation co-benefits, when managed for other ecosystem services provided
by coastal wetlands, such as storm surge attenuation, coastal erosion control, habitat for commercially important species, and ecotourism (Howard et al. 2017). Climate change mitigation refers to efforts to reduce the negative impacts of anthropogenic climate change by either reducing carbon dioxide ($CO_2$) and other greenhouse gas emissions or enhancing natural carbon sinks to increase the rate at which $CO_2$ is removed from the atmosphere. Climate change mitigation for blue carbon involves limiting habitat destruction by human activity, because blue carbon ecosystems naturally store accumulated carbon in their soils for
centuries or millennia (Duarte et al. 2005). When the ecosystem is degraded, the stored carbon can be released, and annual uptake of carbon by the ecosystem ceases (McLeod et al. 2011, Pendleton et al 2012). To better inform policies that identify priority areas for conservation, more precise measurement of carbon stocks and accumulation potential are needed (Howard et al. 2017).

Global estimates of salt marsh area, carbon stocks, and carbon accumulation rates (CAR) are subject to large
uncertainties. Duarte et al. (2013) noted a 20-fold uncertainty in global estimates of salt marsh area, ranging from 22,000-to 400,000 $km^2$. This uncertainty is attributed to ambiguous classification schemes for wetlands. For example, some classification systems consider freshwater and saltwater marshes in the same category (Duarte et al. 2013). Similarly, the estimated, global soil carbon stock of all salt marshes ranges between 0.4 and 6.5 Pg C, a 16-fold range (Duarte et al. 2013).

Currently, the average global CAR estimate for salt marshes is 244.7 ± 26.1 g C $m^{-2}$ $yr^{-1}$ (Ouyang and Lee 2014), but
recent reviews of salt marsh CAR estimates disproportionately represent certain areas of the world (Ouyang and Lee 2014; Chmura 2003). Some areas, such as Europe and eastern North America, have dozens of CAR data points, while others, such as western North America, East Asia, and Australia, have fewer than 10 estimates per region. Regions such as Africa, India, and South America have no data. Furthermore, the high variability in CAR from site to site combined with the 20-fold uncertainty in global marsh area estimates result in global salt marsh CAR estimates ranging from 0.9 to 31.4 Tg C $yr^{-1}$ (Ouyang
and Lee 2014). This 35-fold range is 7 times greater than the global range for mangroves (Ouyang and Lee 2014; Donato et al. 2011). Thus, quantification of the role of salt marshes in the global carbon cycle remains uncertain, and without further sampling from understudied regions, global estimates cannot yet be assumed to reflect the true global carbon sequestration value of salt marshes, leaving an incomplete picture of their importance for greenhouse gas mitigation.




An additional factor that limits CAR quantification is the extensive use of [137]Cs radioisotope dating or a marker horizon method, which have the potential for producing overestimates of sediment accumulation rates when compared to radioisotope dating methods such as [210]Pb (e.g. Callaway et al. 2012; Johannessen and MacDonald 2016). For example, of 143 studies reviewed by Ouyang and Lee (2014), only three did not use either a [137]Cs or marker horizon method, not including studies which did not specify. Dating using these methods have been demonstrated to produce CAR estimates up to 26 % higher than [210]Pb in California salt marshes (Callaway et al. 2012).

The Commission for Environmental Cooperation (CEC), a tri-national governmental organization promoting scientific cooperation between Canada, the United States, and Mexico, identified the Pacific coast of Canada as a significant blue carbon data gap. Additionally, a review of global salt marsh CAR data identified only eight sites on the entire Pacific coast of the continent, none of which were north of 38.2 °N (Ouyang and Lee 2014). This study aims to address this data gap by providing CAR and carbon stock measurements from the Pacific Coast of Canada as a part of the government of Canada's contribution to a continent-wide assessment of blue carbon mitigation potential. We sampled seven salt marshes within the United Nations Educational, Scientific, and Cultural Organization (UNESCO) Clayoquot Sound Biosphere Reserve, British Columbia's Tofino Mudflats Wildlife Management Area, and Pacific Rim National Park Reserve of Canada on Vancouver Island, British Columbia. We calculated soil carbon density (SCD) from dry bulk density (DBD) and percent carbon (% C) on sediment cores collected across the high and low marsh zones of each marsh and used [210]Pb dating in a subset of these cores to quantify carbon accumulation rates. We then used aerial imagery to estimate the extent of high marsh and low marsh areas and estimate carbon stocks and total annual carbon accumulation for each marsh studied. Finally, we place these new data within the context of CAR estimates from salt marshes from the Pacific coast of the United States, Canada's eastern coast, and in Northern Europe. For greenhouse gas mitigation and accounting purposes, we note the importance of methane emissions from wetlands with low salinities (IPCC 2013). We attempted to choose sites with salinity >5 where such emissions are low enough to result in net carbon sequestration (IPCC 2013), but otherwise we focus solely on soil carbon storage and accumulation.

## 2. Methods

### 2.1 Study Area

Clayoquot Sound is a sparsely populated inlet on the west coast of Vancouver Island, British Columbia, Canada, and consists of many islands and peninsulas within mountainous topography. Clayoquot Sound is home to several protected area designations, including Long Beach Unit of Pacific Rim National Park Reserve of Canada, the Province of British Columbia's Tofino Mudflats Wildlife Management Area, and the UNESCO Clayoquot Sound Biosphere Reserve, which protects 366,000 hectares of the west coast of Vancouver Island (Fig. 1). The region is part of the temperate rainforest biome with high annual





rainfall (3270 mm y$^{-1}$) and average annual temperature of 9.5 °C (Environment Canada, 1981-2010 averages). The mean tidal range in Tofino is 2.14 m (Fisheries and Oceans 2016).

We collected 34 sediment cores from seven marshes during Summer, 2016, to determine their carbon storage and accumulation rates (Fig. 1; Appendix A1). These marshes include: (1) Cannery Bay East (CBE), a 4 ha marsh surrounding

5   Kenn Falls Creek, immediately north of the Kennedy River mouth; (2) Cannery Bay West (CBW), a 0.51 ha marsh at the mouth of a creek flowing south into Kennedy Cove, (3) Cypress River Flats (CRF), a 27.42 ha tidal marsh and mud flat, partially within two Indian Reserves of the Ahousaht Nation and immediately north of the Cypress River mouth; (4) Grice Bay at Kootowis Creek (GBK), an 11.69 ha salt marsh located in southeast Grice Bay; (5) Kennedy Cove South (KCS), a 0.78 ha marsh located at a creek mouth on the south shore of Kennedy Cove; (6) "Shipwreck Cove" (SWC), a 1.02 ha marsh in a cove

10  approximately 2.5 km southwest of Kennedy Cove; and (7) Tofino Mudflats (TMF), a 1.5 ha marsh at a creek mouth within the Tofino Mudflats Wildlife Management Area. These sites were identified as typical of salt marshes along Canada's Pacific coast because they include small, pocket marshes encompassing an enclosed, semi-circular area of coastline as well as larger, estuarine marshes. Surface water salinity ranged from 5.9 at KCS to 24 in Grice Bay, and 29 at Roberts Point 6 km south of CRF (Postlethwaite and McGowan 2016, submitted).





**Figure 1-** Study area and marsh locations shown within Clayoquot Sound on the west coast of Vancouver Island (inset), British Columbia, Canada. Site locations clockwise from upper left: Cypress River Flats (CRF), Cannery Bay East (CBE), Cannery Bay West (CBW), Kennedy Cove South (KCS), Shipwreck Cove (SWC), Grice Bay at Kootowis Creek (GBK), and Tofino Mudflats (TMF). Pacific Rim National Park and Reserve (blue crosshatching) covers the southern portion of the map and the Tofino Mudflats Wildlife Management Area (pink crosshatching) covers portions of the southwestern area. The entire region lies within the Clayoquot Sound UNESCO Biosphere Reserve (purple outline, see inset). Tide and climate measurements were recorded at the town of Tofino (orange dot).

**2.2 Field Sampling**

Within each marsh, sediment cores were extracted along linear transects perpendicular to the low tide shoreline following the methodology of Howard et al. (2014). Coring spots were approximately evenly spaced along the transect (between nine and 24 meters apart) from land to sea and spanned the low and high marsh zones (Chmura et al. 2011). Core



locations were chosen to avoid ditches and channels without organic soil accumulation which comprised a relatively small portion of total marsh surface area (see Sect. 4.4.2).

Vegetation composition was recorded as an indicator of 'low' vs 'high' marsh zones. A 50 x 50 cm quadrat was placed over each coring spot, the overhead view was photographed, and the species composition noted. Coring spots were considered low marsh if the species *Triglochin maritima*, *Salicornia spp.*, *Fucus ssp.* or *Ditschilis spicata* were present. A coring spot would be considered high marsh if it included *Plantago maritima, Deschampsia caespitosa, Grindelia integrifolia, Potentilla anserina, Glaux maritima,* or *Eleocharis ssp*. If a spot contained a mixture of these species, the majority percent cover of high or low marsh species was used to determine whether the spot was low or high marsh. *Carex lyngbyei* were often found throughout both strata and so were not considered unique to one zone. These designations are defined by the presence or absence of low marsh or high marsh vegetation-- particularly the high marsh plants *Grindelia integrifolia* and *Potentilla anserina*, which grow in a narrow elevation range in Clayoquot Sound (Jefferson 1973). The high marsh species' ranges align approximately with the mean extreme high-water line of estuarine marshes in Clayoquot Sound, while low marsh encompasses elevations between the mean lower high water and the mean extreme high-water lines (Jefferson 1973 as cited in Deur 2000; Weinmann et al. 1984). This method was groundtruthed using detrended correspondence analysis to verify that vegetation assemblages reflected distinct low and high marsh zones (Hill and Gauch 1980; see Appendix B).

Sediment cores were collected using a simple percussion coring technique in which a length of two-inch (57 mm) diameter, PVC vacuum tubing fitted with a plastic core catcher (AMS Inc.) was hammered into the ground until the depth of refusal. Depth of refusal (DoR) is considered a reasonable proxy for sampling to the maximum depth of organic accumulation (Fourqurean et al. 2014b). At one site (GBK) a steel sledge corer (AMS Inc.) was used to extract four cores, but mechanical problems required switching to the simpler method described above. All cores were stored upright between sampling until their return to the laboratory where they were photographed, logged, and stored under refrigeration at a Parks Canada laboratory in Vancouver, British Columbia.

## 2.3 Estimating Marsh Areas

ArcMap 10.3 tools were used with 50 x 50 cm resolution aerial orthophotos taken in July 2014 (Government of British Columbia) to obtain area estimates of high and low marsh zones. The difference between high marsh and low marsh was delineated by eye between darker-green, denser high marsh vegetation and lighter-green, salt-tolerant, and less-dense low marsh vegetation. This method was groundtruthed using the detrended correspondence analysis (e.g. Hill and Gauch 1980) of vegetation survey data and was found to accurately categorize 94 % of the cores into the correct marsh zone (see Appendix B).

## 2.4 Soil Carbon Density and Carbon Stocks

For each marsh, average carbon stocks (Mg C ha$^{-1}$) were estimated in each sediment core by first measuring the soil carbon density (SCD, g C cm$^{-3}$, Eq. (1)) on one-cm thick sample intervals over the length of each core. SCD is the mass of





carbon found in a cubic centimetre of soil at a given depth and is the product of the organic carbon content % C and the dry bulk density (DBD):

$$SCD \left(\frac{g\,C}{cm^3}\right) = \left(\frac{\%C}{100}\right) \times DBD \qquad (1)$$

where DBD represents the weight of one cc volume of soil that was dried for no less than 72 hours at 60°C.

Organic carbon content (%C) was estimated either using loss-on-ignition (LOI, Eq. (2)) or using CN Elemental and coulometric analysis (Froehlich 1980). An LOI test was performed on every 1 cm subsample by homogenizing samples with a mortar and pestle, combusting them at 550°C for four hours, weighing, and combusting again at 1000°C for two hours (Heiri

et al. 2001). The percentage mass loss-on-ignition (%LOI) was estimated as:

$$\% \, LOI = (DW_i - DW_f)/DW_i * 100 \qquad (2)$$

where $DW_i$ is initial dry weight and $DW_f$ is the dry weight after burning. The %C was also estimated by measuring total carbon (%TC) and inorganic carbon (%IC) on a subset of 93 samples. %TC was measured on these homogenized subsamples using

dry combustion elemental analysis with an Elementar Elemental Analyzer for CN analysis at the University of British Columbia's Department of Earth, Ocean, and Atmospheric Sciences. The same subsamples were then analyzed for %IC using a UIC CM5014 $CO_2$ coulometer connected to a UIC CM5130 acidification module in the Climate, Oceans, and Paleo-Environments (COPE) laboratory at Simon Fraser University. Measurements of %IC were subtracted from the %TC measurements to estimate %C (Hodgson and Spooner 2016; Hedges and Stern 1984; Schumacher 2002; Howard et al. 2014).

Inorganic carbon was negligible in all 93 of the subsamples analysed (max: 0.015 %) and assumed to be zero for all carbon calculation purposes. The relationship between %LOI and %C for these 93 samples was then used to convert %LOI to %C for all sediment samples (Eq. (3), see Appendix C, Fig. C1):

$$\%C = 0.44(\%LOI) - 1.80 \qquad (3)$$

Next, the carbon stock of a core was estimated from the sum of all 1-cm intervals in each core (Eq 4):

$$C\,stock_{core}(g\,C\,cm^{-2}) = \sum_{i=0}^{n} SCD_i \times 1\,cm \qquad (4)$$





Where $i$ = the depth of the top of a 1-cm subsection in cm, $n$ = the depth of the top of the deepest subsection of the core (cm), and $SCD_i$ = the SCD of each subsection $i$ in grams C cm$^{-3}$.

Carbon stocks were calculated both in megagrams per hectare (Mg C ha$^{-1}$) -- the typical unit used in carbon stock assessment (Fourqurean et al. 2014a) -- and in total Mg C for high and low marsh to compare the estimates for each marsh zone. First, to calculate the average carbon stock for all marshes in megagrams C per hectare, all core C stock estimates were averaged across each marsh and scaled up:

$$C\ stock_{Marsh}\ (Mg\ C\ ha^{-1}) = \left( \frac{1}{x} \times \sum_{i=1}^{x} C\ stock_{core} \right) \tag{5}$$

Where x = the number of cores in a marsh.

A Kruskal-Wallis test of significance for sample groups of unequal variances was used to test for significant differences between $C\ stock_{Marsh}$ (Mg C ha$^{-1}$) between the seven marshes studied. Lastly, the Clayoquot Sound average C stock, $C\ stock_{CS,}$ was computed by averaging the C $stock_{marsh}$ estimates from all seven marshes.

Characteristics for low and high marshes were estimated and compared, including total C stock, DBD, %C, SCD, and DoR (Welch's t test). Lastly, the total C stock for low marsh $C\ stock_{LowCS}$ was estimated by averaging each site's low marsh core C stock estimates and multiplying by the total estimated low marsh area in Clayoquot Sound. The same was done to estimate the total high marsh C stock, $C\ stock_{HighCS}$.

**2.5 Carbon Accumulation Rate**

Carbon accumulation rates (CARs) were estimated in five cores from the CBE, CRF, GBK (2), and TMF sites, by multiplying the sediment accumulation rates (SAR) by the SCD (Eq. (6)):

$$CAR\ \left( \frac{g\ C}{m^2 yr} \right) = SAR\ \left( \frac{m}{yr} \right) \times SCD\ \left( \frac{g\ C}{m^3} \right) \tag{6}$$

SARs were calculated from age models determined using $^{210}$Pb dating. Subsamples from each of the five cores were dated using Polonium-210 alpha counting by Core Scientific International (Winnipeg, Canada) and MyCore Scientific (Dunrobin, Canada). Using a constant rate of supply model, age-depth models were constructed, and SARs estimated (Oldfield and Appleby 1984; Rowan et al. 1994; see Appendix C). Some core compaction (maximum 40 %) occurred during the coring process, which would affect our estimated accumulation rates. We corrected for this compaction by applying a correction factor for each core (Eq. (7)):





$$correction\ factor = \frac{recovered\ core\ length\ (cm)}{length\ of\ core\ penetration\ (cm)} \qquad (7)$$

and used it to find the uncompacted depth (Eq. (8)) of any given subsample (Fourqurean et al. 2014a):

$$uncompacted\ depth = sample\ depth\ (cm) \times correction\ factor \qquad (8)$$

The uncompacted depths were used only to calculate SAR (cm yr$^{-1}$), which was then used to calculate CAR (see equation 6).

The regional average CAR in from Clayoquot Sound, $CAR_{CS}$, was calculated as the average of all five cores with $^{210}$Pb dating. The total CAR for a marsh with a dated core was calculated by multiplying the high marsh core CAR times the

high marsh area. Low marsh CAR for each site used the one low marsh dated core multiplied by the site's low marsh area. Regional average CAR for the high and low marsh zones specifically were estimated using the average of the four, $^{210}$Pb dated high marsh cores to represent the high marsh and the one low marsh core to represent the low marsh zone.

### 3.  Results

### 3.1 Soil Properties

Depths of refusal ranged from five cm in the low marsh of SWC to a maximum of 60 cm in the high marsh of CBE. With few exceptions, marsh soils in Clayoquot Sound consisted of three layers separated by defined horizons: topsoil, peat, and sand/clay layers. In all cores, organic carbon concentrations were highest in the surface layers (10-45 %) and decreased to lowest values (~ 2 %) in the deepest parts of the cores (Fig. 2). Soil carbon densities averaged 0.037 ± 0.17 g C cm$^{-3}$ for all sites, and site-wide average SCDs ranged from 0.020 to 0.055 g C cm$^{-3}$ (Appendix A, Table A1). With few exceptions, SCDs

remained relatively constant in the upper parts of the cores and decreased towards the base of the cores where lowest % C values were encountered (Fig. 3).

**Figure 2 Percent Carbon (vertical axis) by depth in centimetres (horizontal axis) for all cores, divided by site. Lightest grey cores are closest to the shoreline while darkest grey cores are furthest.**





**Figure 3 Soil Carbon Density in grams C cm-3 (vertical axis) by depth in centimetres (horizontal axis) for all cores, divided by site. Lightest grey cores are closest to the shoreline while darkest grey cores are furthest.**

**3.2 Carbon Storage and Marsh Area**

The seven marshes ranged in size from 0.51 to 27.42 ha, with a total area of 46.93 ha (Table 1). The high marsh made up 19-63 % of each individual marsh and 58 % (27.39 of 46.94 ha) of the seven marshes we sampled.





The average $C\ stock_{CS}$ for the seven salt marshes is $80.6 \pm 43.8$ Mg C ha$^{-1}$, ranging from $34.6 \pm 22.1$ Mg C ha$^{-1}$ at KCS to $113 \pm 30$ Mg C ha$^{-1}$ at CRF (Appendix A, Table A1). The average $C\ stock_{LowCS}$ is $53.8 \pm 23.0$ Mg C ha$^{-1}$, based on 16 cores from the low marsh zone. The average $C\ stock_{HighCS}$ is $94.9 \pm 28.0$ Mg C ha$^{-1}$, based on 18 cores (Table 1; Fig. 5).

Using our estimates of marsh area, we calculate that $C\ stock_{CS}$ is $4709 \pm 136$ Mg C, 70 % of which is stored in the high marsh.

**INSERT TABLE 1 HERE (Marsh area, carbon stocks, and accumulation rates.)**

### 3.3 Carbon Accumulation Rates

Carbon accumulation rates averaged $146 \pm 102$ g C m$^{-2}$ yr$^{-1}$ at the four sites from which $^{210}$Pb dating was completed.

The low marsh core at GBK had the lowest CAR of 37 g C m$^{-2}$ yr$^{-1}$. CAR in the four high marsh cores ranged from 75 g C m$^{-2}$ yr$^{-1}$ at TMF to 264 g C m$^{-2}$ yr$^{-1}$ at CBE (Fig. 5). The SAR ranged from 0.142 cm yr$^{-1}$ at the GBK low marsh to 1.322 cm yr$^{-1}$ at GBK high marsh (Appendix C Table C1).

Using the CAR from GBK's low marsh as the proxy for all low marsh CAR and the average of the four high marsh cores to estimate the high marsh CAR, we estimate that $CAR_{CS}$ is $54.78 \pm 22.58$ Mg C yr$^{-1}$. Of this, the 19.54 ha of low marsh

accumulate $7.18 \pm 6.24$ Mg C yr$^{-1}$, and the 27.39 ha of high marsh accumulate $47.6 \pm 21.7$ Mg C yr$^{-1}$. Approximately 87 % of the total, annual CAR is in the high marsh, while this area represents only 58% of the total marsh area.

### 3.4 Comparisons between marshes and strata

$C\ stock_{HighCS}$ is significantly higher than $C\ stock_{LowCS}$ ($p < 0.05$) and is largely attributable to differences in the DoR between high and low marshes (Fig. 5). While the average DoR of high marsh cores is significantly higher than the average

DoR of low marsh cores ($p < 0.05$), no significant differences were found between average DBD, average % C, or a core's average SCD in high versus low marsh cores ($p > 0.05$) (Fig. 5).

The Kruskal-Wallis test found significant differences between each of the seven $C\ stock_{Marsh}$ estimates ($p < 0.05$; K = 12.67). This result shows that each of the marsh average carbon stock estimates vary enough from one another that a single site average cannot be assumed to represent the average carbon stocks of all marshes in the region.





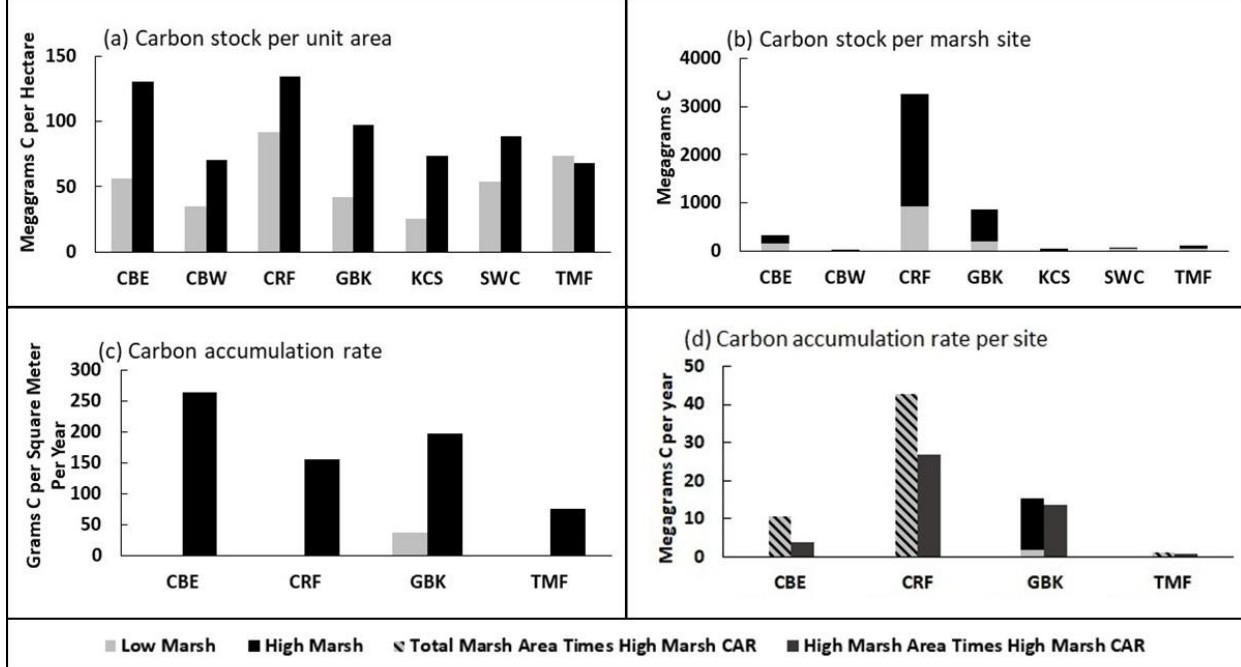

**Figure 4 Carbon stocks and accumulation rates from Clayoquot Sound: (a) stock per hectare (Mg); (b) stock per marsh site (Mg C); (c) Carbon accumulation rates; (d) Annual carbon accumulation for each marsh zone and as contributions to total. In Fig. 4(d), known high marsh and low marsh CAR are used to calculate the total, annual CAR for GBK. For the other sites, the high marsh CAR is extrapolated to the entire marsh area (crosshatched column) and calculated only for the high marsh area (dark gray).**



**Figure 5 Comparison of high marsh and low marsh soil characteristics across all cores in Clayoquot Sound. Only DoR and carbon stock estimates were significantly different (Welch's t test, p < 0.05). high marsh n=18, low marsh n=16 (graphs a, b, c, e, & f). For graph (d), n=7 marshes.**



## 4. Discussion

### 4.1 Carbon Stocks- Comparisons

The *C stock$_{CS}$* averaged $80.6 \pm 43.8$ Mg C ha$^{-1}$, which is roughly half the global estimate of 162 Mg C ha$^{-1}$ made for the top meter of salt marsh soils (Duarte et al. 2013). These global estimates are computed from samples to a depth of one m, while cores from Clayoquot Sound averaged 26.7 cm to DoR. While additional carbon might be stored in deeper, fossil layers below the DoR in Clayoquot Sound marshes, such as in a layer of paleosols buried by tsunami deposits approximately 300 years ago (Clague and Bobrowsky 1994), the large reduction in %C observed at or near the DoR suggests that most soil carbon is found above the DoR.

The shallower depth of accumulation, as approximated by DoR, is likely the main driver of the lower carbon stocks of Clayoquot Sound marshes compared with the global average. The Clayoquot Sound stocks are comparable to those of three natural marsh sites in Everett, Washington, USA, which range from 71.7 to 98.5 Mg C ha$^{-1}$ (Crooks et al. 2014). This region experiences a similar climate to Clayoquot Sound and lies within the same latitude band. Likewise, carbon stocks from marshes in Everett are estimated for the top 30 cm of the marsh soils, which is comparable to the average DoR of 26.7 cm at Clayoquot Sound. At the same time, the 0.037 g C cm$^{-3}$ average SCD of Clayoquot Sound is close to the average of 0.030 g C cm$^{-3}$ calculated from eight National Estuarine Research Reserves in the United States, which includes a site in San Francisco Bay with median SCD of approximately 0.040 g C cm$^{-3}$ (Grimes and Smith 2016). Thus, the carbon stocks in Clayoquot Sound are lower than global averages because high-carbon soil accumulation occurs over depths substantially shallower than one meter.

### 4.2 Carbon Accumulation Rates- Regional applicability and World Comparisons

We anticipate that the CAR results we obtained are likely applicable to mesotidal estuarine (tidal range 2-4 m; Kirwan and Guntersbergen 2010) and pocket marshes throughout the west coast of Vancouver Island and potentially throughout the coast of British Columbia, including an area of up to 60 square kilometers (Ryder et al. 2007). Additionally, the organic carbon values we encountered in peat and sand (ranging from 0-48 %C) layers are similar to those found in previous studies of paleosediments in salt marshes both within Pacific Rim National Park Reserve and from the Ucluelet peninsula approximately 30 km to the south (ranging from 12-62 %C) (Clague and Bobrowsky 1994), suggesting that soil carbon content in British Columbia salt marshes does not vary substantially over short distances.

While the Clayoquot Sound regional average CAR of 146 g C m$^{-2}$ yr$^{-1}$ appears lower than the global average of 245 g C m$^{-2}$ yr$^{-1}$, this difference is not statistically significant (Welch's t-test, $p > 0.05$). The lack of significance is due to high intra-site variability in both the Clayoquot Sound and the global average datasets. Similarly, Clayoquot Sound's average CAR is comparable to CAR estimates from both its latitude band and the other sites within its biogeographical region (Ouyang and Lee, 2014). Clayoquot Sound's average CAR is comparable to the median value for the 48.4-58.4° N range (153.5 g C m$^{-2}$ yr$^{-1}$) and not statistically different from the latitudinal average of 315 g C m$^{-2}$ yr$^{-1}$ (which is skewed by two high CAR estimates of 793 and 1133 g C m$^{-2}$ yr$^{-1}$ (Andrews et al. 2008)). If we consider slightly different subsets of the latitudinal data, the average



CARs from Clayoquot Sound are also not significantly different from the average CARs for the Atlantic coast of North America or Northern Europe (Welch's t test, p > 0.05). This includes the NW Atlantic region (172.2 g C m$^{-2}$ yr$^{-1}$; n=64; 35.0-47.4 °N), and the subset of the NW Atlantic region in Atlantic Canada which is closer in latitude to Clayoquot Sound (188 g C m$^{-2}$ yr$^{-1}$; n=40; 43.6-47.4 °N) (Ouyang and Lee 2014). At the same time, the average Clayoquot Sound CAR is also not

significantly different (Welch's t test, p < 0.05) from the NE Pacific average of 174 g C m$^{-2}$ yr$^{-1}$ (SEM ± 45.1 g C m$^{-2}$ yr$^{-1}$), which were estimated from eight data points in California, USA (Ouyang and Lee 2014). These results both underscore that while there is site-to-site variability in CAR, on the scale of 10-degree latitude bands or biogeographical region, Clayoquot Sound's average CAR is close to expected values for its region and its latitude.

### 4.3 $^{210}$Pb and $^{137}$Cs Dating

Previous researchers have argued that using $^{137}$Cs dating to establish age models can result in SARs, and therefore also CARs, that are biased high (Johannessen and MacDonald 2016). This inflation can occur because $^{137}$Cs dating relies on comparing radionuclide concentrations down the core in relation to the peak atmospheric concentration in 1963, which can result in overestimates due to post-depositional soil turbation (Johannessen and MacDonald 2016). Marker horizons can also be subjected to the same post-depositional soil turbation. Specifically, dating methods using $^{137}$Cs have been shown to produce

slightly - but not significantly- higher CAR estimates in salt marsh CAR estimates, with CAR calculated with $^{210}$Pb an average of 26 % lower (SAR 29 % lower) than the same sites dated using $^{137}$Cs (Callaway et al. 2012).

This overestimation can be a point of concern when making global estimates of salt marsh CAR because the dating method may artificially elevate estimated carbon sequestration potential. All but three of the accumulation rate estimates from the NE Pacific, NW Atlantic, and the 48.4-58.4 °N latitude band in the Ouyang and Lee (2014) compilation were generated

using either $^{137}$Cs dating or a marker horizon method. Given our use of $^{210}$Pb dating to generate CAR estimates, it is possible that the Clayoquot Sound CAR, which is slightly but not significantly lower than other regions of North America and its latitude band, is closer to the true average for both the 48.4-58.4° N latitude range and the NE Pacific biogeographical region. However, our results and those of Ouyang and Lee (2014) suggest that, within the current datasets, spatial heterogeneity in CAR is substantially larger than the differences between dating techniques.



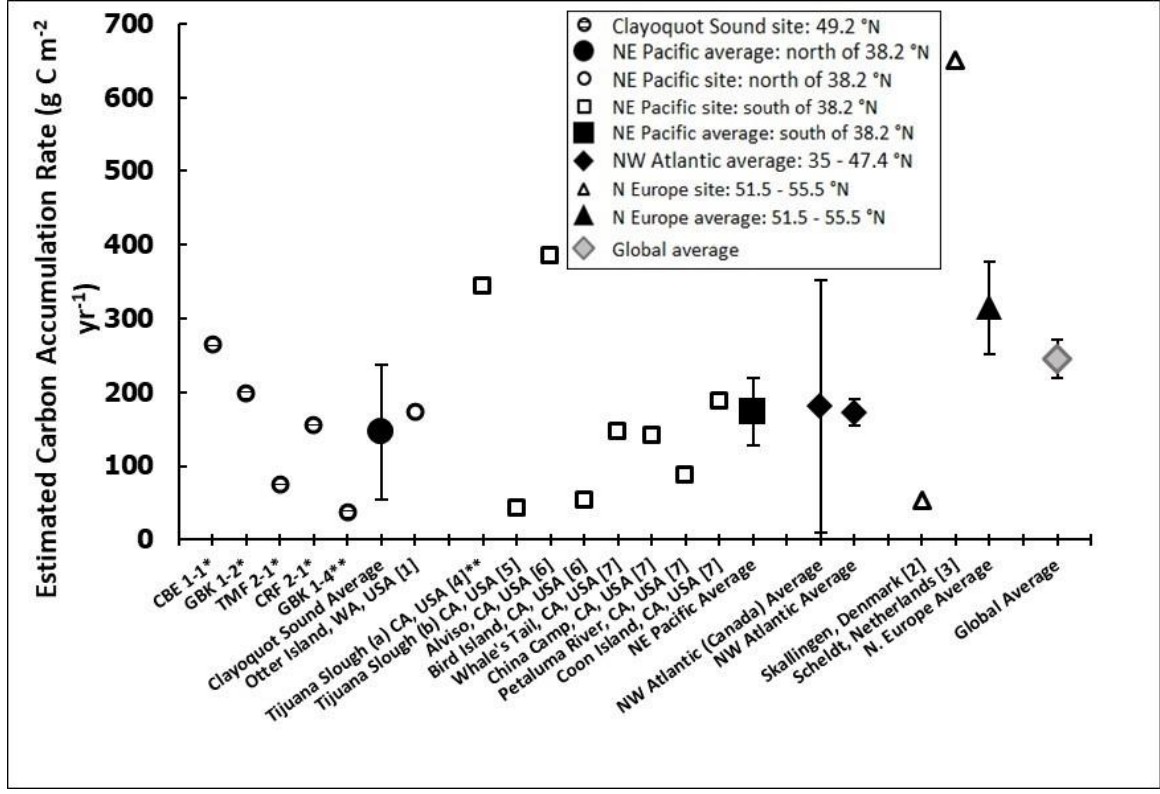

**Figure 6- Comparison of Clayoquot Sound CAR with other salt marsh studies compiled by Ouyang and Lee (2014) grouped by regions as defined by that study.**

All eight available data points from the NE Pacific region south of 38.2 °N are shown; North Europe data points are the minimum (Skallingen, Denmark) and maximum of that dataset (Scheldt, Netherlands). Data from single sites are unfilled shapes, while filled-in shapes represent averages. *= high marsh; **= low marsh; No asterisks= not specified.

[1] Crooks et al. 2014; [2] Callaway et al. 1996; [3] Oenema and Delaune 1988; [4] Cahoon et al. 1996; [5] Chmura et al. 2003; [6] Patrick and Delaune 1990; [7] Callaway et al. 2012. All regional averages aside Clayoquot Sound's, global average, and region definitions from Ouyang and Lee 2014.

## 4.4 Low Marsh CAR

While previous studies have found that low marsh CARs are consistently higher than CARs from high marsh areas (Adams et al. 2012; Callaway et al. 1996; Connor et al. 2001; Elsey-Quirk et al. 2011), our results show that CARs were significantly *lower* in the low marsh at GBK when compared with both the high marsh at GBK, and with the high marsh cores from the other sites. We suggest two factors that may have contributed to this result. First, falling relative sea level (RSL) in Clayoquot Sound may influence marsh accretion dynamics. Low marshes accumulate inorganic sediment primarily from tidal inundation, as particles fall out of suspension in the water or become trapped by the roots of low marsh vegetation (e.g. Connor et al. 2001). Salt marshes thus accumulate vertically in response to rising sea levels (Morris et al. 2002). The tide gauge at Tofino has measured a steadily falling relative sea level since observations began in 1905 (NOAA 2013), which is most likely




a consequence of tectonic uplift in the region (Mazzotti et al. 2008). Therefore, the mechanism of vertical accretion may be different from that observed in marshes experiencing rising sea level. Second, while our one low marsh core exhibits anomalously low CAR relative to the high cores in which CAR was estimated, the single core does not possess sufficient statistical power to draw conclusions about differences between average low and high marsh CAR in Clayoquot Sound because

of regional heterogeneity SAR and CAR within and between marshes. Previous studies of marsh accretion dynamics have demonstrated variability in SAR on scales as small as one meter due to such influences as recent ecological disturbance (Webb et al. 2013), water table height and soil drainage (Craft 2007), and variable mineral sediment deposition from freshwater drainage (Callaway et al. 2012). In our case, a power analysis suggests that at least nine cores measured for CAR would be required to confidently compare the means of low marsh and high marsh cores. This was beyond the resources of our study,

but future studies should consider this to investigate whether the low marsh CAR in Clayoquot Sound is consistently lower than CAR estimates from high marsh areas.

### 4.5 Implications- Clayoquot Sound Salt Marsh vs. Canada's Boreal Forest

The relevance of blue carbon storage potential to climate change mitigation depends on the scale over which it is considered. Per unit area, blue carbon ecosystems have been estimated to accumulate carbon at rates that are substantially

higher than the soils of terrestrial forests (McLeod et al. 2011). A first-order comparison for Canadian ecosystems shows the same pattern of higher carbon uptake rates per unit area in tidal wetlands when compared with net ecosystem productivity of terrestrial forests. Forested areas in parts of Canada are estimated to take up carbon at rates ranging from 35 g C $m^{-2}$ $yr^{-1}$ (Canada-wide estimate, Stinson et al., 2011) to 63 g C $m^{-2}$ $yr^{-1}$ (British Columbia, Peng et al., 2014) while the CAR for salt marshes in Clayoquot Sound are two to three times higher, averaging $146 \pm 102$ g C $m^{-2}$ $yr^{-1}$. (Note that for our comparison

we considered the estimates of net ecosystem productivity, which are higher than C uptake rates of $4.6 \pm 2.1$ g C $m^{-2}$ $yr^{-1}$ for boreal forest soils cited by McLeod et al., 2011). However, consideration of the areal extent of each of these ecosystems is useful for placing the role of blue carbon uptake into perspective for Canadian ecosystems. Estimates of total salt marsh area in Canada range from 44,000 ha (Bridgham et al., 2006) to 111,274 ha (Mcowen et al., 2017). In contrast, the boreal forest ecosystem is one of Canada's largest terrestrial biomes and encompasses approximately 270 million ha (Kurz et al. 2013). If

the CAR estimate from Clayoquot Sound is assumed to approximate the average for all tidal salt marshes in Canada (see Sect. 4.2 for comparison with Eastern Canada marsh sites), Canada's marshes would accumulate between 19,400 and 276,000 Mg C $yr^{-1}$. This is between 0.1-4.1 % of the 6,750,000-18,090,000 Mg C accumulated annually by boreal forests.

An important component of this calculation is that this blue carbon accumulation occurs annually in only 0.016--0.1 % as much land area, which has implications for how blue carbon sequestration potential may be most useful. On a global,

national, and even provincial scale, total amounts of blue carbon sequestration in salt marshes are relatively small compared to large areal expanses of forest. However, blue carbon sequestration in salt marshes is very high per unit area. Hence salt marsh carbon accumulation and storage is important at regional scales in coastal areas and should be considered as a co-benefit to activities such as ecological restoration and conservation for habitat, biodiversity and coastal resilience.

**Conclusion**

Our work provides estimates of soil carbon stocks and accumulation rates from salt marshes on the Pacific coast of Canada, addressing the data gap within North American blue carbon identified by the CEC. The results show that carbon stocks are lower than the global average but are close to values from Everett, USA nearby in the NE Pacific region. This lower

carbon stock is most likely due to shallow depths of accumulation rather than any property of the soil itself. Soil CAR in Clayoquot Sound is also not significantly different ($p < 0.05$) from the global average, other studies in the NE Pacific region, the Atlantic coast of Canada, Northern Europe, and the 48.4-58.4 °N latitude band. This information should be of value for future first-order estimates of carbon in the northern part of the NE Pacific region, as it provides evidence that carbon stock and CAR are indistinguishable from other sites in this region as well as at similar latitude on the Atlantic coast.

We found lower carbon stock in the low marsh and an anomalously low CAR in the low marsh when compared with the high marsh, however we cannot determine if this CAR result is representative of the region's low marsh in general because it comes from only a single core. Soil properties such as SCD, DBD, and %C are not statistically different between marsh elevation zones. The anomalously low CAR may be due to chance because of small-scale variability in SAR, greater soil formation in the high marsh, or other region-specific factors such as a falling RSL may also influence CAR due to differences

in vertical accretion dynamics within areas of emergent coastline.

Despite their relatively small extent, blue carbon ecosystems should be regarded as carbon accumulation 'hot spots,' and the value of salt marsh's relatively high carbon accumulation should be factored into management decision-making.

Lastly, further development of area estimate methods for high and low marsh designations would allow more precise calculations of carbon storage and accumulation in marshes on small scales. This knowledge, alongside greater understanding

of any variability between high and low marsh CAR, would help to inform the role of blue carbon in both local ecosystem services management and the larger global carbon cycle.

**Appendices**

**Appendix A- Summary Core Information**

**Table A1 Core information for samples from Clayoquot Sound.**

**Appendix B- Groundtruthing- Detrended Correspondence Analysis**

Some inaccuracy was expected when ground-truthing the area estimation method using vegetation surveys, but this

was minor. This approach to differentiating high and low marsh matched with vegetation data for 32 of 34 (94 %) of cores.

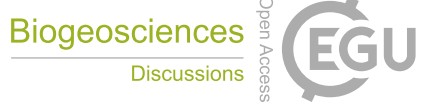

Both CRF 1-2 and CRF 2-2 were classified as low marsh by vegetation survey but fell within the high marsh using the visual orthophotography method. These cores lie 16 m (CRF 1-2) and 12 m (CRF 2-2) away from the boundary with low marsh as measured using orthophotos, which is less than their distances from the nearest high marsh cores (17 m and 23 m, respectively). All other cores fell within the correct marsh zone.

5    A detrended correspondence analysis (Hill and Gauch 1980) of the accuracy of vegetation data showed a reasonably accurate fit of low marsh cores with low marsh vegetation and high marsh cores with high marsh vegetation, plus the addition of a somewhat indistinct, third cluster of vegetation possibly representing the backshore. The classification of marsh strata by presence/absence and percent cover of low marsh or high marsh vegetation was groundtruthed using Canoco v4.5 software. This square root-transformed model accounted for 33.2% of all variance in the vegetation dataset (sum of eigenvalues = 3.29).

10   Cores with low marsh vegetation clustered together while high marsh cores clustered separately. An additional, slightly distinct third cluster of backshore vegetation indicates that some high marsh cores may have been extracted from close to the boundary with a freshwater-dominated backshore or salt-tolerant meadow. The distinction between a salt marsh and a bordering freshwater area has complicated efforts to classify marshes by salinity (Duarte et al. 2013), but this result shows that clustering of vegetation type corresponds reasonably well with each site's designation as high or low marsh.





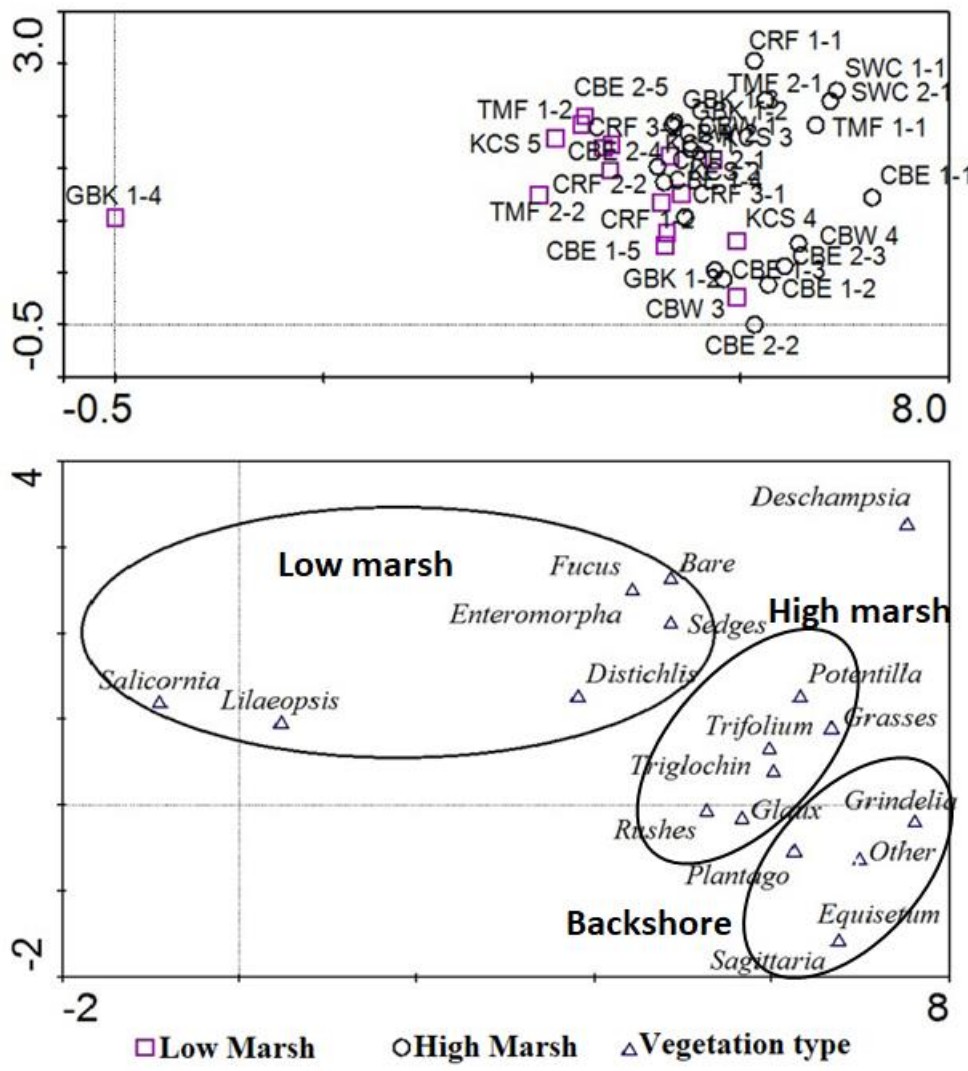

**Figure B1 Detrended correspondence analysis results for Marsh vegetation data. Low marsh cores (top, purple squares) corresponded reasonably well with vegetation identified as low marsh, and high marsh cores corresponded with a distinct cluster of high marsh vegetation. The far bottom-right may indicate a population of less salt-tolerant, backshore vegetation but it is indistinct from the high marsh.**





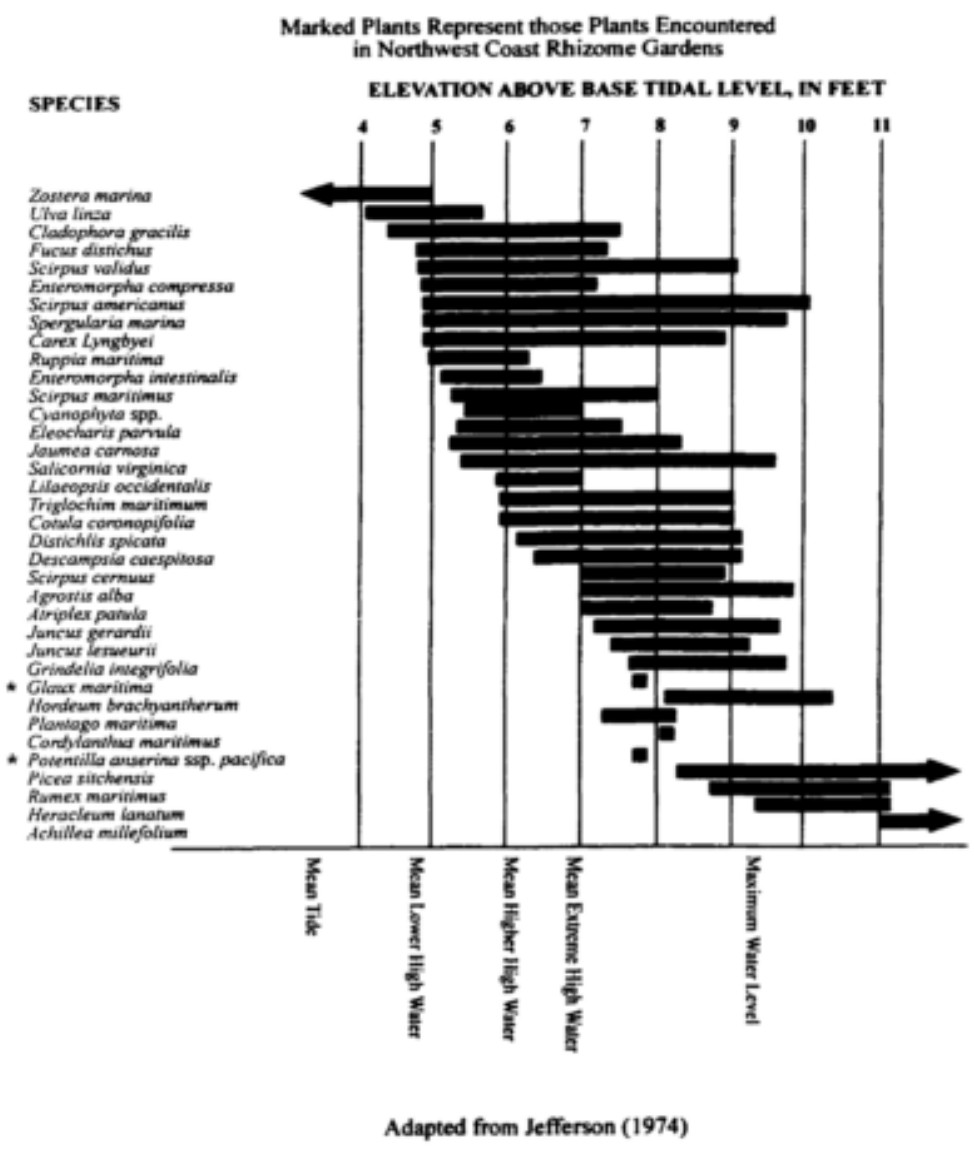

Adapted from Jefferson (1974)

**Figure B2 Reference used for determining marsh stratum based on vegetation. Source: Deur 2000**




**Appendix C- Additional Figures**

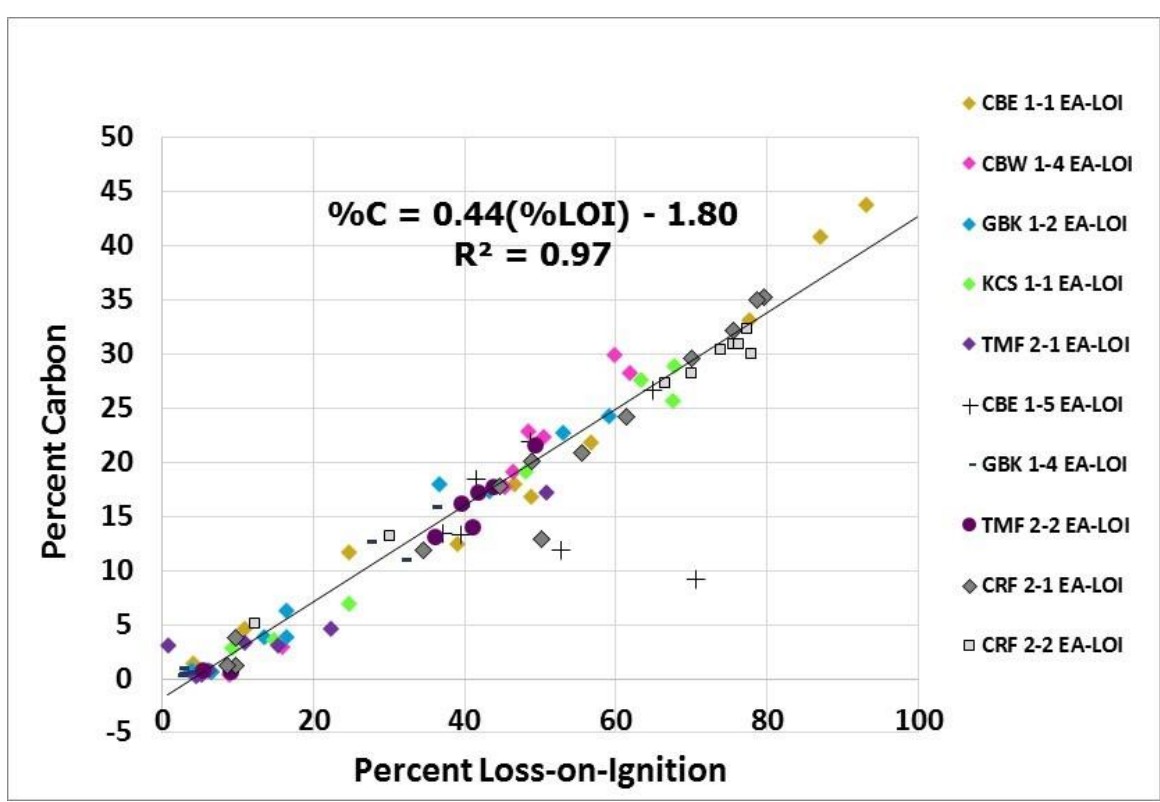

**Figure C1- %LOI-to-%C Relationship. Relationship between measured LOI values and calculated %C, using elemental analyser
EA data on set of 93 subsamples. Measurements from core CBE 1-5 were not used for calculating this relationship due to suspected
5  measurement error.**

**Table C1- Maximum corrected depth of excess $^{210}$Pb activity in centimetres, Age at max $^{210}$Pb depth in years, average vertical, linear
sediment accumulation rate in centimetres per year, average mass accumulation rate in grams per square centimetre per year, and
10  carbon accumulation rate in grams of organic carbon per square meter per year for cores from four different sites.**

**Table C2 $^{210}$Pb data for core CBE 1-1.**



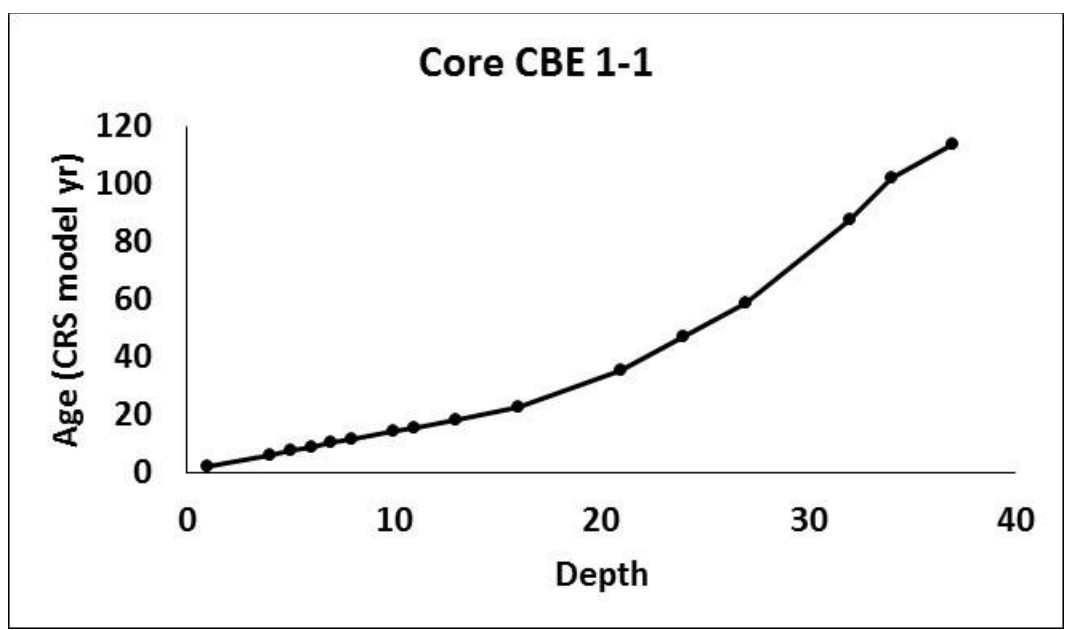

**Figure C2 Core CBE 1-1 Age (yr) vs Depth (cm)**

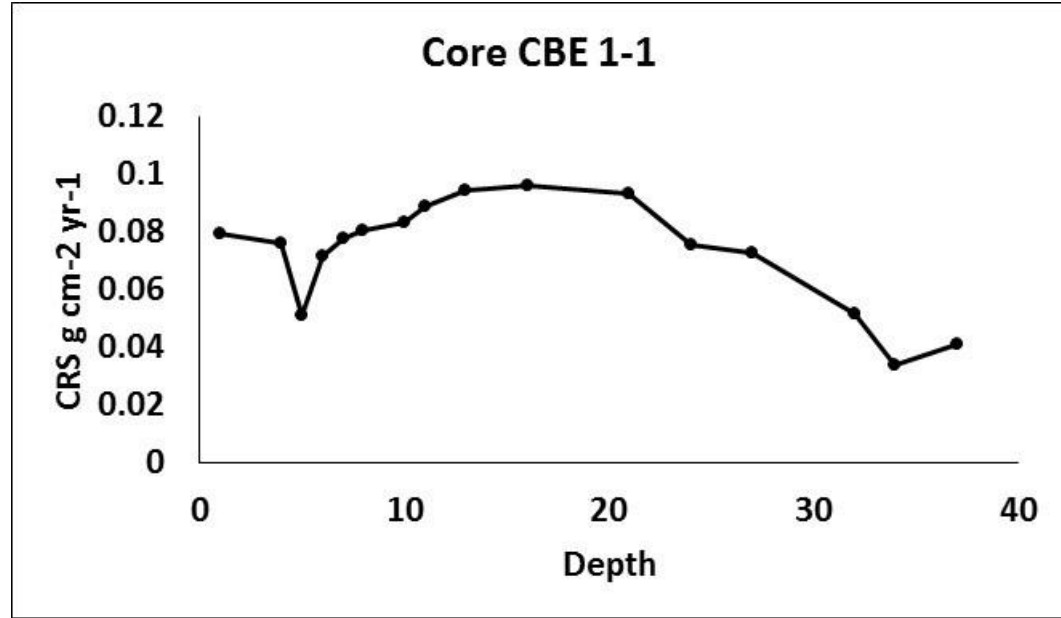

5   **Figure C3 Core CBE 1-1 Sediment Accumulation Rate (CRS g cm$^{-2}$ yr$^{-1}$) vs Depth (cm)**

**Table C3 $^{210}$Pb data for core CRF 2-1.**



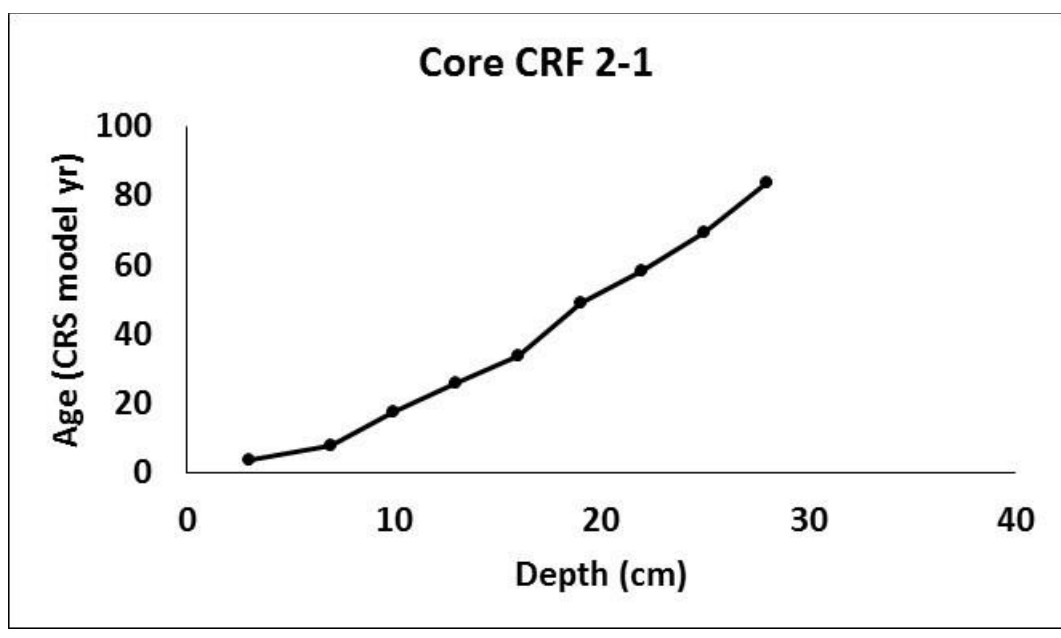

**Figure C4 Core CRF 2-1 Age (yr) vs Depth (cm)**

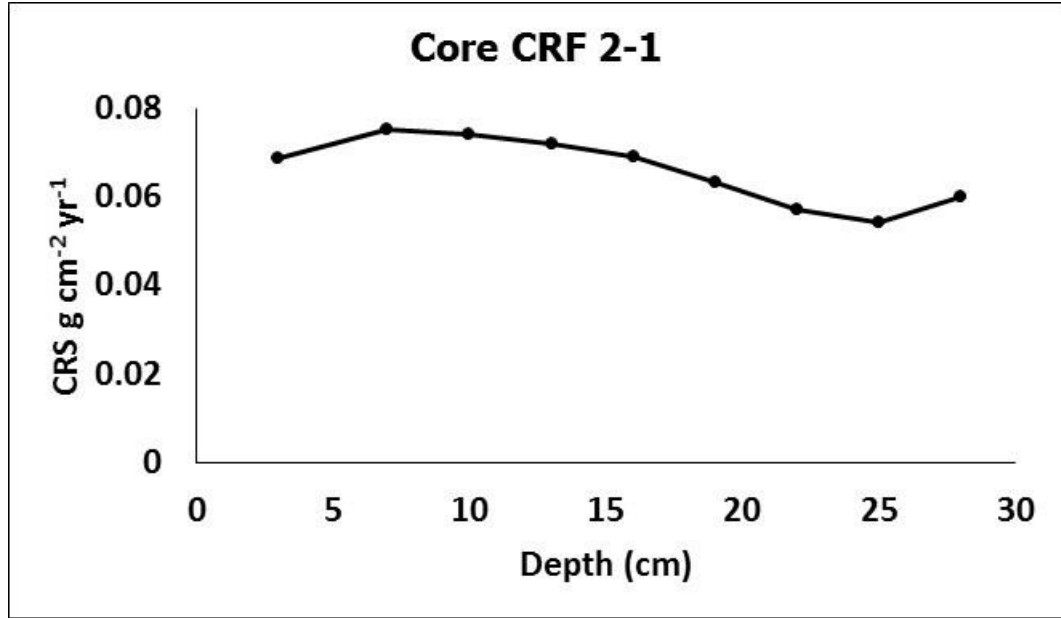

5    **Figure C5 Core CRF 2-1 Sediment Accumulation Rate (CRS g cm$^{-2}$ yr$^{-1}$) vs Depth (cm)**

**Table C4 $^{210}$Pb data for core GBK 1-2.**





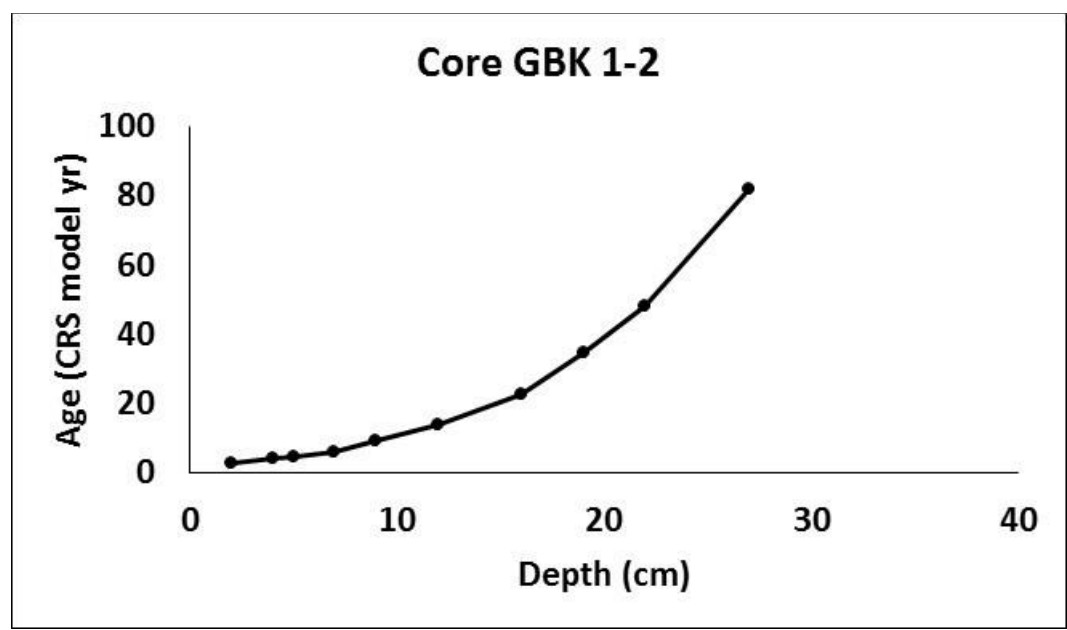

**Figure C6 Core GBK 1-2 Age (yr) vs Depth (cm)**

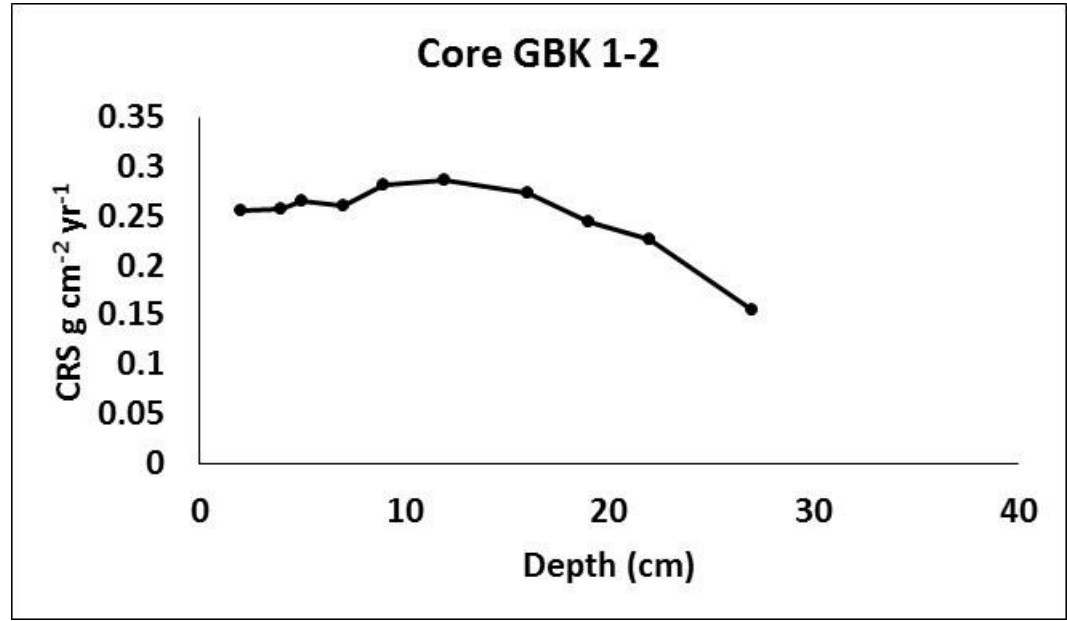

5 **Figure C7 Core GBK 1-2 Sediment Accumulation Rate (CRS g cm$^{-2}$ yr$^{-1}$) vs Depth (cm).**





**Table C5 $^{210}$Pb data for core GBK 1-4.**

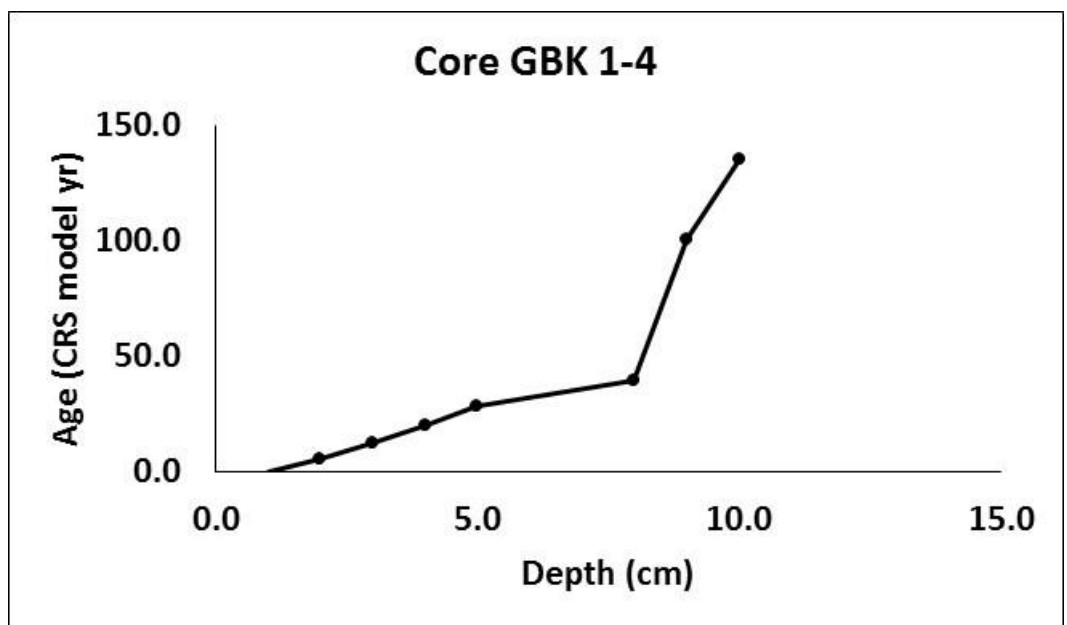

**Figure C8 Core GBK 1-4 Age (yr) vs Depth (cm).**

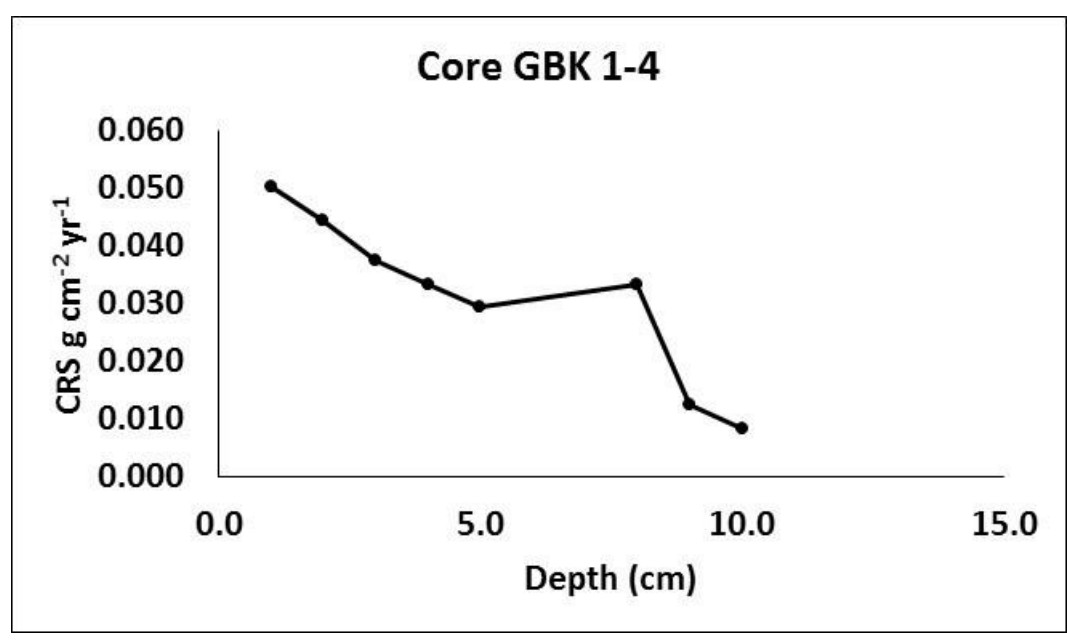

**Figure C9 Core GBK 1-4 Sediment Accumulation Rate (CRS g cm$^{-2}$ yr$^{-1}$) vs Depth (cm)**



**Table C6 $^{210}$Pb data for core TMF 2-1.**

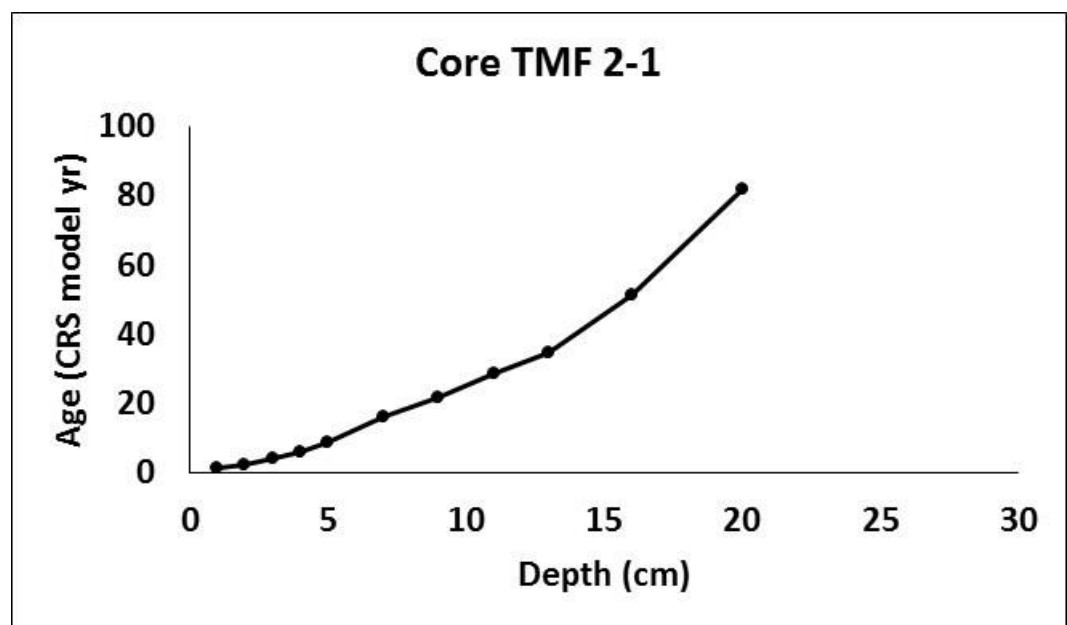

5    **Figure C10 Core TMF 2-1 Age (yr) vs Depth (cm)**

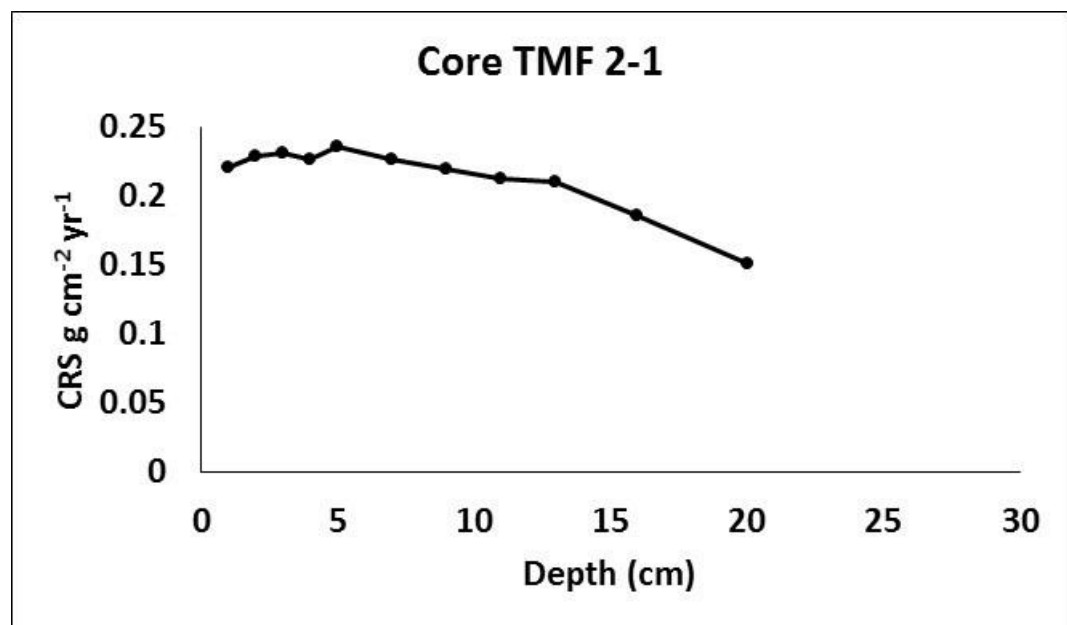

**Figure C11 Core TMF 2-1 Sediment Accumulation Rate (CRS g cm$^{-2}$ yr$^{-1}$) vs Depth (cm)**



## Author Contribution

S. Chastain, M. Pellatt, and K. Kohfeld contributed to experimental design, obtaining external funding, and arranging for fieldwork and in-kind support. S Chastain performed field sample collection, lab analysis, and calculations, K. Kohfeld and M. Pellatt provided guidance and academic supervision to S. Chastain throughout the project: M. Pellatt provided fieldwork guidance and assistance; M. Pellatt and K. Kohfeld provided lab work guidance. S. Chastain, K. Kohfeld, and M. Pellatt co-wrote the manuscript.

## Competing Interests

The authors declare that they have no conflict of interest.

## Disclaimer

## Acknowledgements

The authors extend their gratitude to the following individuals for their contributions to this research, alphabetically: Dr. Richard Atleo, Celeste Barlow, Dr. Douglas Deur, Maija Gailis, Dan Harrison, Hannah Jensen, Victoria Lamothe, Dr. Dana Lepofsky, Gemma MacFarland, Aimee McGowan, Bryn Montgomery, Yiga Phuntsok, Ellie Simpson, Maureen Soon, and Dr. Nancy Turner. We also thank Parks Canada for in-kind support at both Pacific Rim National Park Reserve and the Vancouver office, the Raincoast Education Society for their expertise in the field and access to their grounds to reach our study sites, and the Ahousaht First Nation for allowing us to collect samples from their unceded territory.



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





**Tables**

| Site | Marsh Area (ha) | | | Carbon Stock per hectare (Mg C ha$^{-1}$) | | Marsh Carbon Stock (Mg C) | | | Carbon Accumulation Rate | | | |
|---|---|---|---|---|---|---|---|---|---|---|---|---|
| | | | | | | | | | Per unit area (g C m$^{-2}$ yr$^{-1}$) | | Per marsh (Mg C yr$^{-1}$) | |
| | Low Marsh | High Marsh | Total | Low Marsh | High Marsh | Low Marsh | High Marsh | Total | Low Marsh | High Marsh | Low Marsh | High Marsh |
| Cannery Bay East | 2.57 | 1.42 | 4 | 55.9 | 131 | 144 | 187 | 331 ± 46.3 | - | 264 | - | 10.6 |
| Cannery Bay West | 0.23 | 0.27 | 0.51 | 35.1 | 70.6 | 8.18 | 19.3 | 27.5 ± 26.0 | - | - | - | |
| Cypress River Flats | 10.11 | 17.31 | 27.42 | 92 | 135 | 930 | 2340 | 3270 ± 28.5 | - | 156 | - | 42.8 |
| Grice Bay-Kootowis Creek | 4.79 | 6.9 | 11.69 | 41.6 | 97.3 | 199 | 671 | 870 ± 67.3 | 37 | 198 | 1.72 | 13.7 |
| Kennedy Cove South | 0.32 | 0.47 | 0.78 | 25 | 73.5 | 7.93 | 34.4 | 42.3 ± 73.7 | - | - | - | |
| Shipwreck Cove | 0.83 | 0.2 | 1.02 | 53.6 | 88.7 | 44.3 | 17.6 | 61.8 ± 69.5 | - | - | - | |
| Tofino Mud Flats | 0.69 | 0.82 | 1.51 | 73.7 | 68.1 | 51.1 | 55.7 | 107 ± 7.08 | - | 75 | - | 1.13 |
| **AVERAGE** | - | - | - | 53.8 ± 23.0 | 94.9 ± 28.0 | - | - | - | 37 | 173 ± 79 | - | - |
| **SUM** | 19.54 | 27.39 | 46.93 | - | - | 1385 | 3321 | **4709 ± 136** | - | **-** | **54.78** | |

**Table 1 Marsh area, carbon stocks, and accumulation rates.**





| Site, Core ID | Latitude (49.--- ° N) | Longitude (125.--- ° W) | Marsh stratum (High, Low) | Ave DBD (g cm⁻³) | Ave %C | Ave SCD (g C cm⁻³) | Depth (cm) | C Stock Estimate (Mg C ha⁻¹) |
|---|---|---|---|---|---|---|---|---|
| \multicolumn{9}{c}{Grice Bay Kootowis Creek} |||||||||
| GBK 1-1 | .08754 | .73238 | High | 0.64 ± 0.28 | 6.1 ± 9.8 | 0.016 ± 0.011 | 46 | 73.0 |
| GBK 1-2 | .08756 | .73261 | High | 0.67 ± 0.35 | 5.9 ± 7.8 | 0.016 ± 0.015 | 60 | 60.8 |
| GBK 1-3 | .08763 | .73271 | High | 0.68 ± 0.32 | 6.7 ± 7.2 | 0.027 ± 0.019 | 59 | 158 |
| GBK 1-4 | .08771 | .73283 | Low | 0.66 ± 0.37 | 6.0 ± 6.4 | 0.017 ± 0.017 | 24 | 41.6 |
| Average ± SD | NA | NA | NA | 0.66 ± 0.32 | 6.3 ± 8.0 | 0.020 ± 0.017 | 47.3 ± 16.8 | 83.3 ± 51.4 |
| \multicolumn{9}{c}{Cannery Bay West} |||||||||
| CBW 1-1 | .14115 | .66983 | High | 0.40 ± 0.34 | 15.0 ± 8.3 | 0.033 ± 0.011 | 16 | 46.7 |
| CBW 1-2 | .14115 | .66982 | Low | 0.31 ± 0.12 | 13.4 ± 7.4 | 0.033 ± 0.003 | 7 | 23.5 |
| CBW 1-3 | .14113 | .66962 | Low | 0.17 ± 0.01 | 32.3 ± 7.4 | 0.053 ± 0.012 | 16 | 84.9 |
| CBW 1-4 | .14112 | .66955 | High | 0.19 ± 0.20 | 17.2 ± 8.3 | 0.023 ± 0.012 | 24 | 56.3 |
| Average ± SD | NA | NA | NA | 0.24 ± 0.22 | 20.2 ± 10.7 | 0.035 ± 0.016 | 15.8 ± 7.0 | 52.8 ± 25.4 |
| \multicolumn{9}{c}{Cannery Bay East} |||||||||
| CBE 1-1 | .14139 | .66620 | High | 0.32 ± 0.26 | 19.8 ± 13.1 | 0.036 ± 0.014 | 47 | 169 |
| CBE 1-2 | .14142 | .66629 | High | 0.31 ± 0.17 | 17.0 ± 11.9 | 0.034 ± 0.012 | 38 | 130 |



| | | | | | | | | |
|---|---|---|---|---|---|---|---|---|
| CBE 1-3 | .14147 | .66636 | High | $0.15 \pm 0.06$ | $28.3 \pm 7.7$ | $0.038 \pm 0.004$ | 24 | 90.3 |
| CBE 1-4 | .14152 | .66639 | Low | $0.15 \pm 0.04$ | $23.5 \pm 5.6$ | $0.035 \pm 0.007$ | 14 | 48.4 |
| CBE 1-5 | .14155 | .66644 | Low | $0.23 \pm 0.09$ | $20.2 \pm 5.7$ | $0.048 \pm 0.034$ | 20 | 96.4 |
| CBE 2-2 | .14140 | .66618 | High | $0.16 \pm 0.08$ | $30.3 \pm 5.7$ | $0.045 \pm 0.011$ | 30 | 133 |
| CBE 2-3 | .14142 | .66614 | High | $0.16 \pm 0.05$ | $28.4 \pm 5.4$ | $0.043 \pm 0.004$ | 30 | 132 |
| CBE 2-4 | .14144 | .66609 | Low | $0.53 \pm 0.33$ | $13.6 \pm 13.2$ | $0.035 \pm 0.022$ | 20 | 69.8 |
| CBE 2-5 | .14151 | .66602 | Low | $0.95 \pm 0.23$ | $1.1 \pm 1.2$ | $0.008 \pm 0.008$ | 30 | 9.16 |
| Average ± SD | NA | NA | NA | $0.29 \pm 0.26$ | $21.5 \pm 11.8$ | $0.037 \pm 0.017$ | $28.1 \pm 10.1$ | $97.7 \pm 49.7$ |
| Cypress River Flats | | | | | | | | |
| CRF 1-1 | .27905 | .90754 | High | $0.17 \pm 0.05$ | $26.9 \pm 6.7$ | $0.043 \pm 0.010$ | 38 | 163 |
| CRF 1-2 | .27896 | .90758 | Low | $0.17 \pm 0.04$ | $28.7 \pm 4.1$ | $0.048 \pm 0.009$ | 16 | 76.1 |
| CRF 2-1 | .27935 | .90932 | High | $0.34 \pm 0.31$ | $19.1 \pm 11.0$ | $0.034 \pm 0.014$ | 37 | 126 |
| CRF 2-2 | .27916 | .90926 | Low | $0.20 \pm 0.13$ | $25.3 \pm 9.8$ | $0.040 \pm 0.007$ | 26 | 105 |
| CRF 3-1 | .27890 | .91100 | High | $0.31 \pm 0.29$ | $23.9 \pm 14.3$ | $0.036 \pm 0.013$ | 32 | 117 |
| CRF 3-2 | .27882 | .91087 | Low | $0.29 \pm 0.27$ | $23.3 \pm 13.0$ | $0.041 \pm 0.014$ | 23 | 94.8 |



| | | | | | | | | |
|---|---|---|---|---|---|---|---|---|
| Average ± SD | NA | NA | NA | 0.25 ± 0.23 | 24.1 ± 10.9 | 0.040 ± 0.012 | 28.7 ± 8.6 | 113 ± 30.0 |
| **Kennedy Cove South** | | | | | | | | |
| KCS 1-1 | .13696 | .67082 | High | 0.28 ± 0.20 | 17.9 ± 10.5 | 0.035 ± 0.023 | 24 | 73.5 |
| KCS 1-2 | .13707 | .67085 | High | 0.25 ± 0.17 | 15.0 ± 6.9 | 0.027 ± 0.006 | 14 | 27.4 |
| KCS 1-3 | .13714 | .67093 | High | 0.42 ± 0.32 | 11.1 ± 9.5 | 0.021 ± 0.012 | 16 | 25.6 |
| KCS 1-4 | .13719 | .67096 | High | 0.30 ± 0.27 | 14.7 ± 6.8 | 0.029 ± 0.009 | 10 | 29 |
| KCS 1-5 | .13720 | .67107 | Low | 0.51 ± 0.39 | 7.6 ± 6.8 | 0.018 ± 0.011 | 10 | 17.8 |
| Average ± SD | NA | NA | NA | 0.34 ± 0.28 | 14.0 ± 9.3 | 0.027 ± 0.017 | 14.8 ± 5.8 | 34.6 ± 22.1 |
| **Shipwreck Cove** | | | | | | | | |
| SWC 1-1 | .12995 | .69943 | High | 0.30 ± 0.37 | 25.8 ± 13.8 | 0.031 ± 0.011 | 29 | 43.9 |
| SWC 2-1 | .13014 | .69908 | High | 0.60 ± 0.59 | 21.0 ± 10.1 | 0.074 ± 0.044 | 18 | 133 |
| Average ± SD | NA | NA | NA | 0.47 ± 0.52 | 23.1 ± 12.3 | 0.055 ± 0.040 | 23.5 ± 7.8 | 88.6 ± 63.3 |
| **Tofino Mud Flats** | | | | | | | | |
| TMF 1-1 | .13014 | .88689 | High | 0.54 ± 0.32 | 11.0 ± 12.0 | 0.027 ± 0.012 | 27 | 72.0 |
| TMF 1-2 | .13020 | .88688 | Low | 0.33 ± 0.16 | 10.6 ± 5.4 | 0.028 ± 0.009 | 26 | 70.5 |
| TMF 2-1 | .12989 | .88661 | High | 0.72 ± 0.35 | 5.2 ± 7.2 | 0.022 ± 0.023 | 28 | 64.2 |



| | | | | | | | | |
|---|---|---|---|---|---|---|---|---|
| TMF 2-2 | .13017 | .88665 | Low | 0.43 ± 0.38 | 13.5 ± 7.1 | 0.033 ± 0.012 | 27 | 76.8 |
| Average ± SD | NA | NA | NA | 0.52 ± 0.34 | 9.9 ± 8.8 | 0.027 ± 0.015 | 27.0 ± 0.8 | 70.9 ± 5.2 |
| REGION AVERAGE ± SD | NA | NA | NA | 0.39 ± 0.33 | 17.0 ± 12.4 | 0.037 ± 0.017 | 26.6 ± 12.7 | 80.6 ± 43.8 |

**Table A1- Summary Core Information for Samples from Clayoquot Sound.**





| Core ID | Maximum Uncompacted Depth of $^{210}$Pb Activity (cm) | Age at Max $^{210}$Pb Depth (yr before June 2016) | Average Sediment Accumulation Rate (cm yr$^{-1}$) | Average Mass Accumulation Rate (g cm$^{-2}$ yr$^{-1}$) | Carbon Accumulation Rate (g C m$^{-2}$ yr$^{-1}$) |
|---|---|---|---|---|---|
| CBE 1-1 (High Marsh) | 48.81 | 113.8 | $0.757 \pm 0.187$ | $0.0731 \pm 0.019$ | 264 |
| GBK 1-2 (High Marsh) | 45.00 | 81.9 | $1.322 \pm 0.462$ | $0.251 \pm 0.038$ | 198 |
| GBK 1-4 (Low Marsh) | 9.41 | 135.54 | $0.142 \pm 0.084$ | $0.0312 \pm 0.014$ | 37 |
| TMF 2-1 (High Marsh) | 17.71 | 81.9 | $0.360 \pm 0.161$ | $0.214 \pm 0.025$ | 75 |
| CRF 2-1 (High Marsh) | 34.46 | 83.8 | $0.460 \pm 0.197$ | $0.066 \pm 0.008$ | 156 |
| Average | 31.08 | 99.4 | $0.725 \pm 0.432$ | $0.151 \pm 0.095$ | $146 \pm 102$ |

Table C1- Maximum corrected depth of excess $^{210}$Pb activity in centimetres, Age at max $^{210}$Pb depth in years, average vertical, linear sediment accumulation rate in centimetres per year, average mass accumulation rate in grams per square centimetre per year, and carbon accumulation rate in grams of organic carbon per square meter per year for cores from four different sites. Compaction factor for GBK 1-4 is average of the other 3 from GBK because hole depth was not measured due to infilling after the corer was withdrawn. No
5 significant difference in CAR was found when using the minimum (0 %) and maximum (40 %) compaction from other cores ($p > 0.05$).



| Sample ID | DBD (g cm$^{-3}$) | Upper Depth (cm) | Lower Depth (cm) | Extrapolated Upper Section Depth (cm) | Extrapolated Lower Section Depth (cm) | Age at Bottom of Extrapolated Section (yr) | CRS Sediment Accumulation Rate (g cm$^{-2}$ yr$^{-1}$) |
|---|---|---|---|---|---|---|---|
| 0+1 | 0.17 | 0 | 1 | 0 | 1 | **2.1** | 0.0795 |
| 4 | 0.11 | 1 | 4 | 1 | 4 | **6.3** | 0.076 |
| 5 | 0.09 | 4 | 5 | 4 | 5 | **8** | 0.0511 |
| 6 | 0.08 | 5 | 6 | 5 | 6 | **9** | 0.0719 |
| 7 | 0.1 | 6 | 7 | 6 | 7 | **10.3** | 0.0781 |
| 8 | 0.11 | 7 | 8 | 7 | 8 | **11.6** | 0.0806 |
| 10 | 0.12 | 8 | 10 | 8 | 10 | **14.4** | 0.0835 |
| 11 | 0.11 | 10 | 11 | 10 | 11 | **15.7** | 0.0887 |
| 13 | 0.11 | 11 | 13 | 11 | 13 | **18.1** | 0.0948 |
| 16 | 0.15 | 13 | 16 | 13 | 16 | **22.7** | 0.0963 |
| 21 | 0.24 | 16 | 21 | 16 | 21 | **35.6** | 0.0932 |
| 24 | 0.3 | 21 | 24 | 21 | 24 | **47.4** | 0.0755 |
| 27 | 0.28 | 24 | 27 | 24 | 27 | **58.8** | 0.0729 |
| 32 | 0.3 | 27 | 32 | 27 | 32 | **87.9** | 0.0518 |
| 34 | 0.24 | 32 | 34 | 32 | 34 | **102.2** | 0.0337 |
| 37 | 0.16 | 34 | 37 | 34 | 37 | **113.8** | 0.0411 |
| 42 | 0.78 | 37 | 42 | 37 | 42 | | |
| 46 | 0.92 | 42 | 46 | 42 | 46 | | |

**Table C2 $^{210}$Pb data for core CBE 1-1.**



| Sample ID | DBD (g cm$^{-3}$) | Upper Depth (cm) | Lower Depth (cm) | Extrapolated Upper Section Depth (cm) | Extrapolated Lower Section Depth (cm) | Age at Bottom of Extrapolated Section (yr) | CRS Sediment Accumulation Rate (g cm$^{-2}$ yr$^{-1}$) |
|---|---|---|---|---|---|---|---|
| 0+1+2 | 0.08 | 0 | 2 | 0 | 3 | 3.440742 | 0.0688 |
| 4+5+6 | 0.08 | 4 | 6 | 3 | 7 | 7.854617 | 0.0752 |
| 8+9+10 | 0.24 | 8 | 10 | 7 | 10 | 17.732 | 0.0744 |
| 13 | 0.18 | 10 | 13 | 10 | 13 | 25.74563 | 0.0721 |
| 16 | 0.18 | 13 | 16 | 13 | 16 | 33.75461 | 0.0691 |
| 19 | 0.32 | 16 | 19 | 16 | 19 | 48.95086 | 0.0632 |
| 22 | 0.18 | 19 | 22 | 19 | 22 | 58.15044 | 0.0572 |
| 25 | 0.21 | 22 | 25 | 22 | 25 | 69.56789 | 0.0544 |
| 28 | 0.29 | 25 | 28 | 25 | 28 | 83.7887 | 0.0602 |
| 31 | 0.91 | 28 | 31 | 28 | 31 | | |
| 34 | 1.02 | 31 | 34 | 31 | 34 | | |

**Table C3 $^{210}$Pb data for core CRF 2-1.**




| Sample ID | DBD (g cm$^{-3}$) | Upper Depth (cm) | Lower Depth (cm) | Extrapolated Upper Section Depth (cm) | Extrapolated Lower Section Depth (cm) | Age at Bottom of Extrapolated Section (yr) | CRS Sediment Accumulation Rate (g cm$^{-2}$ yr$^{-1}$) |
|---|---|---|---|---|---|---|---|
| 0+ 2 | 0.3275 | 0 | 2 | 0 | 2 | 2.559908 | 0.255869 |
| 4 | 0.173833 | 2 | 4 | 2 | 4 | 3.910491 | 0.25742 |
| 5 | 0.192733 | 4 | 5 | 4 | 5 | 4.635432 | 0.265861 |
| 7 | 0.2041 | 5 | 7 | 5 | 7 | 6.152198 | 0.260655 |
| 9 | 0.4064 | 7 | 9 | 7 | 9 | 9.037745 | 0.28168 |
| 12 | 0.455 | 9 | 12 | 9 | 12 | 13.80398 | 0.286389 |
| 16 | 0.5945 | 12 | 16 | 12 | 16 | 22.47162 | 0.274354 |
| 19 | 0.997333 | 16 | 19 | 16 | 19 | 34.68199 | 0.245038 |
| 22 | 1.0045 | 19 | 22 | 19 | 22 | 47.96489 | 0.226871 |
| 27 | 1.060833 | 22 | 27 | 22 | 27 | 81.87581 | 0.156415 |
| 33 | 0.8173 | 27 | 33 | 27 | 33 | | |
| 37 | 0.818467 | 33 | 37 | 33 | 37 | | |

**Table C4 $^{210}$Pb data for core GBK 1-2.**





| Sample ID | Depth of Top of Section (cm) | Depth of Bottom of Section (cm) | DBD (g cm⁻³) | Sediment Accumulation Rate (g cm⁻² yr⁻¹) | Age at top of section (yr) |
|---|---|---|---|---|---|
| "GBK 1-4 0cm" | 0.0 | 1.0 | 0.254 | 0.050 | 0.0 |
| | 1.0 | 2.0 | 0.288 | 0.045 | 5 |
| "GBK 1-4 2cm" | 2.0 | 3.0 | 0.248 | 0.038 | 13 |
| | 3.0 | 4.0 | 0.240 | 0.033 | 20 |
| "GBK 1-4 4cm" | 4.0 | 5.0 | 0.283 | 0.030 | 28 |
| | 5.0 | 8.0 | 0.914 | 0.033 | 39 |
| "GBK 1-4 8cm" | 8.0 | 9.0 | 0.265 | 0.012 | 101 |
| | 9.0 | 10.0 | 0.260 | 0.008 | 136 |
| "GBK 1-4 10cm" | 10.0 | 11.0 | 0.411 | | |
| | 11.0 | 12.0 | 0.413 | | |
| GBK 1-4 12cm" | 12.0 | 13.0 | 0.944 | | |
| | 13.0 | 14.0 | 1.120 | | |
| GBK 1-4 14cm" | 14.0 | 15.0 | 0.990 | | |
| | 15.0 | 16.0 | 1.055 | | |
| GBK 1-4 16cm" | 16.0 | 17.0 | 1.028 | | |
| | 17.0 | 18.0 | 0.999 | | |
| GBK 1-4 18cm" | 18.0 | 19.0 | 0.974 | | |
| | 19.0 | 20.0 | 1.067 | | |
| "GBK 1-4 20cm" | 20.0 | 21.0 | 0.961 | | |
| | 21.0 | 22.0 | 1.020 | | |
| "GBK 1-4 bottom" | 22.0 | 23.0 | 1.023 | | |
| "GBK 1-4 bottom" | 23.0 | 24.0 | 1.089 | | |

Table C5 $^{210}$Pb data for core GBK 1-4.





**Table C6 $^{210}$Pb data for core TMF 2-1.**

| Sample ID | DBD (g cm$^{-3}$) | Upper Depth (cm) | Lower Depth (cm) | Extrapolated Upper Section Depth (cm) | Extrapolated Lower Section Depth (cm) | Age at Bottom of Extrapolated Section (yr) | CRS Sediment Accumulation Rate (g cm$^{-2}$ yr$^{-1}$) |
|---|---|---|---|---|---|---|---|
| 0+1 | 0.292 | 0 | 1 | 0 | 1 | 1.323574 | 0.220615 |
| 2 | 0.2191 | 1 | 2 | 1 | 2 | 2.283357 | 0.228281 |
| 3 | 0.4043 | 2 | 3 | 2 | 3 | 4.031063 | 0.231332 |
| 4 | 0.485 | 3 | 4 | 3 | 4 | 6.17555 | 0.226765 |
| 5 | 0.6516 | 4 | 5 | 4 | 5 | 8.943209 | 0.235434 |
| 7 | 0.8063 | 5 | 7 | 5 | 7 | 16.04905 | 0.22694 |
| 9 | 0.6366 | 7 | 9 | 7 | 9 | 21.85414 | 0.219325 |
| 11 | 0.748 | 9 | 11 | 9 | 11 | 28.87923 | 0.212951 |
| 13 | 0.5987 | 11 | 13 | 11 | 13 | 34.57198 | 0.210338 |
| 16 | 1.0538 | 13 | 16 | 13 | 16 | 51.53309 | 0.186391 |
| 20 | 1.1437 | 16 | 20 | 16 | 20 | 81.90333 | 0.150634 |
| 24 | 1.3173 | 20 | 24 | 20 | 24 | | |