# Peer review of "Carbon Stocks and Accumulation Rates in Salt Marshes of the Pacific Coast of Canada"

_Biogeosciences, 2018_

## Referee Comment (RC1) · Anonymous Referee #1 · 1 May 2018

The article submitted by Chastain and al aims to estimate carbon stocks and accumulation rates in salt marshes of the Pacific coast of Canada. The argument is the capacity of tidal salt marshes to sequestrate C. However there are still a limited number of marshes for which carbon accumulation have been estimated. In this work, the authors present an new investigation of salt marshes of the Pacific Canadian coast. The paper address a relevant scientific questions within the scope of BG and present new data. But I have a major problem with the estimate of the mass accumulation rate. There is no details on 210Pb data and not enough on the dating method. Indeed to estimate CAR, it is necessary to estimate carbon but also sediment accumulation rates. It is classical to use 210Pb for dating sediments over the last decades. Measurements by gamma spectrometry permit to determine also 226Ra, the supported 210Pb, and

137Cs, an independent time marker. Here the authors use an another technic, the alpha spectrometry. The problem of this method is that it measures 210Pb only. But the 210Pb-dating method is based on the decay of the excess Pb, ie the fraction of 210Pb not supported by its radioactive parent (226Ra) in sediment. This implies the authors made assumptions to estimate this supported fraction, this information is not given in the article. How the authors determine this supported value ? did they use the same value for the 4 cores. What is the error associated witih the assumption ? In addition, 210Pb/210Pbxs are not presented which is a critical aspect as these data define the SAR. The method used to estimate SAR/MAR is also not enough detailed. From the CSR (constant rate of supply) model) based on the inventories, it is possible to calculate directly age of each layer, and then to estimate SAR and MAR, such values would have been interesting to discuss also (temporal trends, potential change in accretion regarding sea level rise). I do not understand why the authors speech two times about 137Cs, there is useless. In fact in such environnements, where accretions could have change, 210Pb IN EXCESS is indeed appropriate, 137Cs is interesting only to validate the chronology.

The second problem is the sampling. The authors explain "Within each marsh, sediment cores were extracted along linear transects perpendicular to the low tide shoreline following the methodology of Howard et al. (2014). Coring spots were approximately evenly spaced along the transect"(between nine and 24 meters apart) from land to sea and spanned the low and high marsh zones". On the other hand, the authors determine SAR only on 4 cores sampled in different systems. Do they assume there is no change in sediment according to the position along the transects ? What about the morphology along these transect ? Regarding the purpose of the article, I would have expect to have a higher number of cores on which 210Pb was determined in order to obtain more reliable SAR and then CAR. Whereas 210Pb is already mentioned in the abstract, there is no data of this radionuclides nor figures presenting profiles with depth. Considering the objectives of the article, that imply to know rather precisely SAR/MAR in order to calculate CAR, the number of dated cores is also too weak to be

representative of the different systems. I do not recommend publication of this work.

---

## Referee Comment (RC2) · Anonymous Referee #2 · 22 May 2018

General comments: Assessing the organic carbon stocks and accumulation rates of vegetated coastal ecosystems is a topic of interest in recent years, increasing number of papers aim at quantifying the potential of these ecosystems to mitigate $CO_2$ emissions through their management in an approach described as Blue Carbon. However, limitations such as the scarcity of estimates of organic carbon accumulation rates, uncertainties in the area covered by these ecosystems or the large areas still unsampled are precluding their inclusion into existing carbon mitigation strategies. The article by Chastain et al. aims to address a data gap in saltmarsh ecosystems by providing new estimates of organic carbon stocks and accumulation rates in saltmarshes of Clayoquot Sound, in the Pacific Coast of Canada, where no data are available. The scope of the paper is valid- It is important to increase the number of available data on C stocks

and, in particular, on carbon accumulation rates in these ecosystems to enlarge provincial, national, and global databases to finally develop policy priorities for conservation. While the data compilation on organic carbon stocks presented in this manuscript is commendable, insufficient data on carbon accumulation rates (CAR) is provided for the purpose of estimating carbon accumulation rates in the Pacific Coast of Canada. I found major flaws in information regarding the estimation of sediment accumulation rates (SAR), hence CAR. Also, there are a series of miss-points in the methodology used for the estimation of C stocks and accumulation rates (detailed in the comments below) that the authors should take into consideration to achieve the publication of this work.

Specific comments on the estimation of carbon accumulation rates (CAR):

I have major concerns about how carbon accumulation rates (CAR) are estimated. First, authors only estimate CAR in a total of five cores collected at 4 marshes, although they sampled a total of 34 sediment cores for C stock determination. The authors do not explain why only these cores were dated, or whether other cores were also analyzed by 210Pb but could not be dated. Mixing, erosion or changes in sedimentation are common processes in coastal sediments, and could lead to the alteration of sediment records, hence 210Pb concentration profiles (Ruiz-Fernández and Hillaire-Marcel, 2009). However, altered 210Pb profiles, although not datable, are results themselves.

The authors do not report data on total or excess 210Pb specific activities and no explanations are given regarding the determination of supported 210Pb, which might vary between marshes but also along the depth of their sediment profiles, especially if soils consist of three marked layers, topsoil, peat, and sand/clay (section 3.1, line 16-17).

The application of the CRS dating model to estimate SAR is unclear and some arguments should be provided regarding the election and application of this model. To

apply this model certain assumptions must be met, for instance, this model is based in excess 210Pb inventories, which implies that the excess 210Pb horizon should have been reached in all dated cores. Without 210Pb data, this is impossible to evaluate. In addition, the CRS model provides estimates of SAR at each sediment layer rather than average sedimentation rates for the last century. In the main text Chastain et al. report average SARs at each core but do not explain how this average is estimated or if they have normalized SARs to a certain age-depth.

A results section showing 210Pb concentration profiles, 210Pb inventories and esti- mated fluxes should be included in the paper, this is important to evaluate whether the dating model applied is valid and to discuss the uncertainties associated to the estimation of ages and SAR. In the current version of the manuscript the authors in- clude a section comparing 210Pb and 137Cs dating, which I believe is unnecessary; the authors did not analyze 137Cs in their cores and 137Cs is most commonly used to validate 210Pb chronologies. There are many aspects that can bias SAR and CAR high, for instance the presence of sediment mixing in 210Pb concentration profiles. My recommendation to the authors is to look critically at their 210Pb data and discuss the uncertainties related to their age-depth models, SAR and CAR estimates.

Second, to estimate CAR authors use sediment accumulation rates (SAR) which they multiply by the soil carbon density (SCD). While they acknowledge that sediment com- paction occurred during coring and so they correct SAR for potential compaction, they do not correct SCD for such. This might lead to an overestimation of CAR. The authors estimate SCD multiplying the percent carbon content (%C) by the soil dry bulk density (DBD). While the rationale behind this is correct, soil DBD should be corrected for core compaction prior to the estimation of SCD. The mass contained in one cc volume of soil after coring occupies a greater volume in the field (before compaction occurs). Related to core compaction, I disagree with the statement in equation 8 used to estimate the uncompacted depth of a given subsample. Let's assume the recovered core length is 50 cm and the core penetration is 100 cm. This would result in a correction factor of

0.5 following equation 7. Then, if the correction factor is applied in equation 8 and is multiplied by the subsample depth (i.e., 1 cm-thick slice) this do not result in an uncompacted depth, please revise. In addition, compaction would unlikely have been linear throughout the soil column due to the presence of different soil layers (topsoil, peat, and sand/clay), which may show different degrees of compaction. For this reason, any variable that is sensitive to soil compaction such as DBD, SCD and SAR should not be used for the determination of CAR or C stocks. Variables such as the mass depth (m) (or mass per unit of area; g cm-2) and mass accumulation rate (MAR; g cm-2 yr-1) are not affected by soil compaction, then should be used instead of DBD and SAR to avoid the propagation of errors in the determination of CAR or C stocks (see below). My recommendation to the authors is to recalculate CAR as:

CAR (g C m-2 yr-1) = MAR (g cm-2 yr-1) x %C

Where the % C is not the average percentage of C along the sediment column but the fraction of the accumulated mass of C (gC cm-2/ g soil cm-2), estimated from the sum of the sediment layers accumulated over a period t = 100 yr, which should be approximately where the excess 210Pb horizon is reached.

To finish with concerns about CAR estimates, I think differentiation between low and high marsh CAR is not possible with only one CAR estimate for a low marsh. The authors indeed acknowledge this at the end of the manuscript in section 4.4, line 3-4. I believe this should be said upfront. Accordingly, comparisons of Clayoquot Sound CAR with other salt marshes should be based only on high marsh CAR estimates reported at the other study sites. Final recommendation to the authors would be to avoid estimating total CAR for a marsh with only a dated core as the high marsh core CAR times the total marsh area (this is represented as a crosshatched column in figure 4). The latter is probably unlikely according to the results presented: lower C stocks in low marsh cores and low CAR in the single low marsh core.

Specific comments on the estimation of C stocks:

Similarly, estimation of C stocks should be done using soil mass depth (g cm-2) rather than SCD multiplied by the thickness of soil slices, i.e., 1 cm, which is affected by soil compaction. My recommendation to the authors is to recalculate C stocks using the soil mass per unit area (m) rather than the sum of all sections DBD x %C x 1 cm. The soil mass per unit area at each layer is not affected by compaction or by inaccurate slicing. It is estimated by dividing the dry sample mass by the area sampled by the core tube, which is the cross-sectional area of its inner diameter (D), $\pi(D/2)\hat{}2$:

Cstock_core=$\sum(DW/(\pi r\hat{}2))$ x %C

The second problem here is the computation of overall averages when the averaged values are computed over a number of estimates that are different at each site or when the area each marsh represents is not the same. - The mean C stock at a marsh (C stockmarsh) should be calculated as the weighted average of the mean Cstockscore estimated in the low marsh area and the mean of those estimated in the high marsh area, being the weights, the area made by low and high marsh at each individual marsh. - Then, the average C stock of low marshes at Clayoquot Sound (Cstock-LowCS) also should be a weighted average, with weights being the low marsh area of each individual marsh. Same for CstockHighCS.

The authors use the depth of refusal (DoR) as a measure of the maximum depth of organic accumulation. C stocks are then estimated down to this depth (average 27.6 cm) and compared with those of global estimates (which some are estimated down to 1 m and others extrapolated to the same depth, 1 m). The authors conclude the C stocks at Clayoquot Sound are lower than those globally, but this is not a fair comparison. DoR is relative to the equipment being used and to the type of soils, therefore I feel that any comparisons made without standardizing all sites to a certain depth/mass depth (preferably) can be misleading. Rather than extrapolating their measurements to 1 m, the authors could normalize global estimates to 30 cm or perhaps to a certain mass depth, which would be the most consistent to establish comparisons (see Wendt and Hauser, 2013). As well, authors could discuss differences on C stocks based on %C,

DBD and CAR rates found globally and at other regions such as eastern Canada, the Pacific Coast of the United States and Mexico.

Authors should take action on the points listed above and revisit their calculations to provide more consistent estimates of C stocks and CAR. As well, they should discuss their results, perhaps, with more emphasis on C stocks and intra marsh variability (for which they have a good dataset), while presenting CAR results in a more local scale, avoiding upscaling to the Pacific coast of Canada. Instead, I encourage the authors to discuss temporal trends in C accumulation at the dated marshes if 210Pb profiles allow so.

Minor comments:

- Introduction; line 2: "Coastal, vegetated ecosystems, such as eelgrass meadows, mangroves…" why only eelgrass? Haven't the other seagrass species been recognized for their ability to store carbon?

-Section 2.2 line 15: please, add in the text the percentage area of ditches and channels rather than directing the reader to section 4.4.2 which does not exist in the present version of this manuscript.

- In section 2.2 there are some aspects the authors should mention such as the length of the PVC tubes and the technique to slice sediment cores. Whether the cores were cut lengthwise or were sliced using a sediment extruder, are important details since compaction may become greater using the latter.

- Figure 2: In the present version of the manuscript authors evaluate C stocks in high and low marsh sites, while the distance to shore is not really a variable they evaluate. Because of that, my recommendation would be to represent percent C profiles using either two colours/shapes for low and high marsh.

-Figure 3 is repetitive if figure 2 is already presented, in addition figure 3 may be misleading as SCD has not been corrected for compaction.

- Section 3.2 Carbon storage and marsh area: DoR measurements at each core/site are key to understand differences in C stocks within sites, the authors should provide this information either in table 1 or Table A1. Then, since C stocks and DoR are highly connected, my suggestion to the authors is to merge section 3.2 and 3.4.

- Appendix C: please, provide uncertainties associated to SAR estimates.

―――――――――――――――――

---

## Author Comment (AC1) · 30 May 2018

SHORT RESPONSE TO ANONYMOUS REVIEWER #1 ON "Carbon stocks and accumulation rates in salt marshes of the Pacific Coast of Canada" by Chastain et al.

We thank anonymous reviewer 1 for comments which we think will help clarify the paper. Here we provide a brief response detailing how we intend to address reviewer concerns.

**1. Reviewer 1 writes: "...I have a major problem with the estimate of the mass accumulation rate. There is no details on 210Pb data and not enough on the dating method...."**

RESPONSE: (1) A raw data file including 210Pbxs and original 210Po activity was

provided to the PANGAEA data archive for release when this paper is published. The appendix tables C2-C6 provided with the manuscript will now also include measured dry bulk density, % organic carbon, soil carbon density, subsample depth intervals (cm), total 210Po/210Pb activity, SD of total 210Po/Pb activity , 210Pbxs, age (yr), SAR, and CAR. (2) more detailed information about 210Pb methods are provided below.

**2. Reviewer 1 continues: "...the authors use an another technic, the alpha spectrometry. The problem of this method is that it measures 210Pb only. But the 210Pb-dating method is based on the decay of the excess Pb, ie the fraction of 210Pb not supported by its radioactive parent (226Ra) in sediment. This implies the authors made assumptions to estimate this supported fraction, this information is not given in the article. How the authors determine this supported value ? did they use the same value for the 4 cores. What is the error associated witih the assumption ? In addition, 210Pb/210Pbxs are not presented which is a critical aspect as these data define the SAR. The method used to estimate SAR/MAR is also not enough detailed."**

RESPONSE: We thank the reviewer for pointing out this lack of clarity in our description of 210Pb dating methods and will revise the methods section accordingly. Our analysis follows published methods in which 210Po is measured using alpha spectrometry (and 210Pb is calculated from this by assuming secular equilibrium (e.g., Binford, 1990). The assumption is made that no unsupported 210Pb is present at the depth of lowest observed activity. Financial constraints prevented us from also measuring 226Ra, wo the lowest 210Po activity is assumed to estimate background for calculating the 210Pbxs throughout the core. We do not use the same background value to estimate supported 210Pb in all 5 cores, as the lowest observed activity varies from core to core. This methodology is consistent with previously published papers using alpha spectrometric techniques to estimate 210Pb systematics and SARs in sedimentary systems (e.g., Brossier et al., 2014; Chambers et al., 2017; Galka et al., 2017; Greiner et al., 2013, Kolker et al. 2009; Wachnicka et al., 2013).

**3. Reviewer #1 further states: "...From the CSR (constant rate of supply) model)**

based on the inventories, it is possible to calculate directly age of each layer, and then to estimate SAR and MAR, such values would have been interesting to discuss also (temporal trends, potential change in accretion regarding sea level rise). . ..''

RESPONSE: Sediment accumulation rates (SARs) were provided in the supplemental information for all 5 dated cores (figures C2 through C11). Soil carbon density vs depth for each core is also plotted in Figure 3 of the manuscript. We can easily add an additional figure and discussion of CAR vs depth for each dated core. We can also include a more thorough discussion of the average and standard deviation of calculated CAR with depth. All associated data will be included in tables with the revision.

**4. Reviewer #1 states: ". . . I do not understand why the authors speech two times about 137Cs, there is useless. In fact in such environnements, where accretions could have change, 210Pb IN EXCESS is indeed appropriate, 137Cs is interesting only to validate the chronology..''**

RESPONSE: The reviewer rightfully points out that 137Cs has limitations, which we describe in the manuscript. Therefore, we are not certain what the reviewer's concerns are with our mention of 137Cs. Our text describes that most previous estimates of marsh CAR on the Pacific Coast (and the full global dataset) are calculated using 137Cs dating, and that our work provides more accurate accumulation rate estimates using 210Pb at our study sites. If necessary, we will edit the text for clarity.

**5. Reviewer #1 writes: "The second problem is the sampling. . . . the authors determine SAR only on 4 cores sampled in different systems. Do they assume there is no change in sediment according to the position along the transects ? What about the morphology along these transect ? Regarding the purpose of the article, I would have expect to have a higher number of cores on which 210Pb was determined in order to obtain more reliable SAR and then CAR. Whereas 210Pb is already mentioned in the abstract, there is no data of this radionuclides nor figures presenting profiles with depth. Considering the objectives of the article, that imply to know rather precisely**

[Figure]

SAR/MAR in order to calculate CAR, the number of dated cores is also too weak to be representative of the different systems."

RESPONSE: First, as stated for #1, data were supplied to PANGAEA for release at publication and will be supplied in the appendix tables with this manuscript.

Second, we acknowledge the reviewer's concerns regarding the small number of cores dated (although we dated 5, not 4 cores). We agree that marsh dynamics result in high spatial variability, which we observed in the field and is apparent in estimations of variability (standard deviations) within and between marshes cores. Our sampling strategy was intended to capture spatial variability of carbon dynamics between different marshes in the region within the limits of our funding budget. However, a comparison of stratigraphies of cores within the marshes will allow us to make a meaningful comparison of stocks and accumulations between the dated and undated cores. For our revision we will highlight the 210Pb dated cores in Figures 2 and 3 to highlight the how representative our 210Pb dated cores are for each marsh and elaborate on this analysis in the Discussion.

Finally, we were surprised to see a recommendation for rejection based on a small sample size, given that we have doubled the number of studies using 210Pb dating on the west coast of North America. Thus, we believe our results substantially improve quantification of CAR on the Pacific coast of North America.

REFERENCES Binford, M.W.: Calculation and uncertainty analysis of 210Pb dates for PIRLA project lake sediment cores, Journal of Paleolimnology, 3, 253-267, 1990.

Brossier, B. F. Oris, W. Finsinger, H. Asselin, Y. Bergeron, A. A. Ali, Using tree-ring records to calibrate peak detection in fire reconstructions based on sedimentary charcoal records, The Holocene 24(6): 635-645, 2014.

Chambers, F. A. Crowle, J. Daniell, D. Mauquoy, J. McCarroll, N. Sanderson, T. Thom, Ph. Toms, J. Webb, Ascertaining the nature and timing of mire degradation; using

palaeoecology to assest future conservation management in Northern England, AIMS Environmental Science, 4(1): 54-82, 2017.

Gałka, M, Szal, M, Watson, EJ et al. (6 more authors). Vegetation Succession, Carbon Accumulation and Hydrological Change in Subarctic Peatlands, Abisko, Northern Sweden. Permafrost and Periglacial Processes, 28 (4). pp. 589-604. ISSN 1045-6740, 2017.

Greiner, J.T., McGlathery, K.J., Gunnell, J., & McKee, B.A.: Seagrass restoration enhances "blue carbon" sequestration in coastal waters, PloS one, 8, (8), 2013.

Kolker, A. S. et al. High-resolution records of the response of coastal wetland systems to long-term and short-term sea-level variability, Estuarine, Coastal, and Shelf Science, 84(4): 493-508, 2009.

Wachnicka, A., L. S. Collins, E. E. Gaiser, Response of diatom assemblages to 130 years of environmental change in Florida Bay (USA) Journal of Paleolimnology 49 (1): 83-101, 2013.

---

## Author Comment (AC2) · 4 Jun 2018

SHORT RESPONSE TO ANONYMOUS REVIEWER #2 ON "Carbon stocks and accumulation rates in salt marshes of the Pacific Coast of Canada" by Chastain et al.

We thank anonymous reviewer #2 for constructive comments which we think will greatly clarify the paper. We have gone through these suggestions, agree with them, and believe that they can be addressed during the revision process. Here we provide a brief response detailing how we intend to address them.

REVIEWER 2: "While the data compilation on organic carbon stocks presented in this manuscript is commendable, insufficient data on carbon accumulation rates (CAR) is provided for the purpose of estimating carbon accumulation rates in the Pacific Coast

of Canada. I found major flaws in information regarding the estimation of sediment accumulation rates (SAR), hence CAR. Also, there are a series of miss-points in the methodology used for the estimation of C stocks and accumulation rates (detailed in the comments below) that the authors should take into consideration to achieve the publication of this work."

**1. REVIEWER 2: "I have major concerns about how carbon accumulation rates (CAR) are estimated. First, authors only estimate CAR in a total of five cores collected at 4 marshes, although they sampled a total of 34 sediment cores for C stock determination. The authors do not explain why only these cores were dated, or whether other cores were also analyzed by 210Pb but could not be dated. Mixing, erosion or changes in sedimentation are common processes in coastal sediments, and could lead to the alteration of sediment records, hence 210Pb concentration profiles (Ruiz-Fernández and Hillaire-Marcel, 2009). However, altered 210Pb profiles, although not datable, are results themselves."**

RESPONSE: Reviewer #2's concern is similar to concerns of Reviewer 1, regarding the number of cores for which 210Pb dates were provided. We dated as many cores as we could, given our project funding limitations. We examined stratigraphies to choose cores that were representative of all cores/sediments from a given marsh. We will do the following in our revision to address this concern: (1) modify Figures 2 and 3 to indicate which cores were dated (2) add a discussion of stratigraphic comparison, and how this uncertainty influences our result (3) point out in our text that we saw no visible or measurable indication of sediment alteration via mixing or erosion in our cores. (4) clarify in the text that we have presented all 210Pb profiles for our cores in the appendix, and none exhibits such alteration.

**2. REVIEWER 2: "The authors do not report data on total or excess 210Pb specific activities and no explanations are given regarding the determination of supported 210Pb, which might vary between marshes but also along the depth of their sediment profiles, especially if soils consist of three marked layers, topsoil, peat, and sand/clay**

(section 3.1, line16-17)."

RESPONSE: As mentioned for Reviewer 1, we will expand our description of methods and provide all data as an appendix.

**3. REVIEWER 2: "The application of the CRS dating model to estimate SAR is unclear and some arguments should be provided regarding the election and application of this model. To apply this model certain assumptions must be met, for instance, this model is based in excess 210Pb inventories, which implies that the excess 210Pb horizon should have been reached in all dated cores. Without 210Pb data, this is impossible to evaluate. In addition, the CRS model provides estimates of SAR at each sediment layer rather than average sedimentation rates for the last century. In the main text Chastain et al. report average SARs at each core but do not explain how this average is estimated or if they have normalized SARs to a certain age-depth."**

RESPONSE: As mentioned for Reviewer 1, we will expand our description of methods and provide all data as an appendix.

**4. REVIEWER 2: "A results section showing 210Pb concentration profiles, 210Pb inventories and estimated fluxes should be included in the paper, this is important to evaluate whether the dating model applied is valid and to discuss the uncertainties associated to the estimation of ages and SAR. This will be included in the revised manuscript."**

RESPONSE: (1) The 210Pb concentration profiles will be included and 210Pb data provided in the paper appendix; (2) SARs based on the CRS model were already provided in appendix files but will be moved to the paper). (3) A more complete discussion of uncertainties of ages and SAR will be included.

**5. REVIEWER 2: "In the current version of the manuscript the authors include a section comparing 210Pb and 137Cs dating, which I believe is unnecessary; the authors did not analyze 137Cs in their cores and 137Cs is most commonly used to validate**

210Pb chronologies. There are many aspects that can bias SAR and CAR high, for instance the presence of sediment mixing in 210Pb concentration profiles. My recommendation to the authors is to look critically at their 210Pb data and discuss the uncertainties related to their age-depth models, SAR and CAR estimates."

RESPONSE: Reviewer #1 had a similar concern with this section of our manuscript. We will revise for clarity and generalize to describe the "many aspects" that can bias SAR and CAR rather than an exclusive focus on 137Cs. For example, we could include a more detailed explanation of the effect of spatial variability, which can affect carbon measurement even on small scales within a single marsh.

**6. REVIEWER 2: "Second, to estimate CAR authors use sediment accumulation rates (SAR) which they multiply by the soil carbon density (SCD). While they acknowledge that sediment compaction occurred during coring and so they correct SAR for potential compaction, they do not correct SCD for such. This might lead to an overestimation of CAR. The authors estimate SCD multiplying the percent carbon content (%C) by the soil dry bulk density (DBD). While the rationale behind this is correct, soil DBD should be corrected for core compaction prior to the estimation of SCD. The mass contained in one cc volume of soil after coring occupies a greater volume in the field (before compaction occurs). Related to core compaction, I disagree with the statement in equation 8 used to estimate the uncompacted depth of a given subsample. Let's assume the recovered core length is 50 cm and the core penetration is 100 cm. This would result in a correction factor of 0.5 following equation 7. Then, if the correction factor is applied in equation 8 and is multiplied by the subsample depth (i.e., 1 cm-thick slice) this do not result in an uncompacted depth, please revise. In addition, compaction would unlikely have been linear throughout the soil column due to the presence of different soil layers (topsoil, peat, and sand/clay), which may show different degrees of compaction. For this reason, any variable that is sensitive to soil compaction such as DBD, SCD and SAR should not be used for the determination of CAR or C stocks. Variables such as the mass depth (m) (or mass per unit of area; g cm-2) and**

[Figure]

mass accumulation rate (MAR; g cm-2 yr-1) are not affected by soil compaction, then should be used instead of DBD and SAR to avoid the propagation of errors in the determination of CAR or C stocks (see below). My recommendation to the authors is to recalculate CAR as:

CAR (g C m-2 yr-1) = MAR (g cm-2 yr-1) x %C

Where the % C is not the average percentage of C along the sediment column but the fraction of the accumulated mass of C (gC cm-2/ g soil cm-2), estimated from the sum of the sediment layers accumulated over a period t = 100 yr, which should be approximately where the excess 210Pb horizon is reached."

RESPONSE: We thank the reviewer for pointing out this error in our assumptions regarding the compaction factor. In our revision we will redo the CAR and Stock estimates without the compaction factor, following the suggestion above.

**7 REVIEWER 2: "To finish with concerns about CAR estimates, I think differentiation between low and high marsh CAR is not possible with only one CAR estimate for a low marsh. The authors indeed acknowledge this at the end of the manuscript in section 4.4, line 3-4. I believe this should be said upfront. Accordingly, comparisons of Clayoquot Sound CAR with other salt marshes should be based only on high marsh CAR estimates reported at the other study sites. Final recommendation to the authors would be to avoid estimating total CAR for a marsh with only a dated core as the high marsh core CAR times the total marsh area (this is represented as a crosshatched column in figure 4). The latter is probably unlikely according to the results presented: lower C stocks in low marsh cores and low CAR in the single low marsh core."**

RESPONSE: We agree and will eliminate discussion of comparison between high and low marsh cores. Figure 4 will be altered to remove the high marsh CAR * total marsh area column.

**8 REVIEWER 2: "Specific comments on the estimation of C stocks: Similarly, estimation of C stocks should be done using soil mass depth (g cm-2) rather than SCD multiplied by the thickness of soil slices, i.e., 1 cm, which is affected by soil compaction. My recommendation to the authors is to recalculate C stocks using the soil mass per unit area (m) rather than the sum of all sections DBD x %C x 1 cm. The soil mass per unit area at each layer is not affected by compaction or by inaccurate slicing. It is estimated by dividing the dry sample mass by the area sampled by the core tube, which is the cross-sectional area of its inner diameter (D), _(D/2)ËĘ2: Cstock_core=P (DW=(_rËĘ2 )) x %C" RESPONSE: This is a very helpful suggestion and we will apply this modification in the revised version of the manuscript.**

**9. REVIEWER 2: "The second problem here is the computation of overall averages when the averaged values are computed over a number of estimates that are different at each site or when the area each marsh represents is not the same. - The mean C stock at a marsh (Cstockmarsh) should be calculated as the weighted average of the mean Cstockscore estimated in the low marsh area and the mean of those estimated in the high marsh area, being the weights, the area made by low and high marsh at each individual marsh. - Then, the average C stock of low marshes at Clayoquot Sound (Cstock-LowCS) also should be a weighted average, with weights being the low marsh area of each individual marsh. Same for CstockHighCS."**

RESPONSE: We agree with this assessment of how to calculate the mean C stocks, and have done this in the manuscript. We will clarify this description, as we separately calculated both the average stock per hectare and the total C stock of the marshes.

**10. REVIEWER 2: "The authors use the depth of refusal (DoR) as a measure of the maximum depth of organic accumulation. C stocks are then estimated down to this depth (average 27.6 cm) and compared with those of global estimates (which some are estimated down to 1 m and others extrapolated to the same depth, 1 m). The authors conclude the C stocks at Clayoquot Sound are lower than those globally, but this is not a fair comparison. DoR is relative to the equipment being used and to the type of soils, therefore I feel that any comparisons made without standardizing**

all sites to a certain depth/mass depth (preferably) can be misleading. Rather than extrapolating their measurements to 1 m, the authors could normalize global estimates to 30 cm or perhaps to a certain mass depth, which would be the most consistent to establish comparisons (see Wendt and Hauser, 2013). As well, authors could discuss differences on C stocks based on %C, DBD and CAR rates found globally and at other regions such as eastern Canada, the Pacific Coast of the United States and Mexico. "

RESPONSE: (1) We agree in principle that it is not appropriate to compare marsh accumulation to a depth of 27.6 cm with a depth of 1m. However, we should be clear that our depth of refusal was based on substrate at the base of the core (gravel), not that we could not core any deeper due to mechanical issues. We will make this point clearer in our Methods and in our Discussion of comparisons with global sites. (2) We can extend our comparison between %C, DBD, SCD, and CAR, although we believe stock comparisons between marshes can be challenging because of an incomplete understanding of the total soil volume, differing methods to estimate this volume, and small-scale variability in soil strata. As the stratigraphies in Clayoquot Sound are relatively simple, we could compare marsh carbon stocks with other studies to a depth of 30 cm, with appropriate caveats added to clarify these uncertainties.

**11.REVIEWER 2: "Authors should take action on the points listed above and revisit their calculations to provide more consistent estimates of C stocks and CAR. As well, they should discuss their results, perhaps, with more emphasis on C stocks and intra marsh variability (for which they have a good dataset), while presenting CAR results in a more local scale, avoiding upscaling to the Pacific coast of Canada. Instead, I encourage the authors to discuss temporal trends in C accumulation at the dated marshes if 210Pb profiles allow so. "**

RESPONSE: We thank Reviewer 2 for these very helpful comments. In our revision we intend to revisit these calculations and document how changing them affects our conclusions. Although we will continue to place our results within the context of global estimates, we will also emphasize the role of local intra-marsh variability and how stock

characteristics (DBD, %C, SCD) compare locally and regionally. As stated for our short response to reviewer 1, we will discuss temporal trends in C accumulation.

**12. Minor Comments RESPONSE: We have reviewed these minor requests and will implement them in our revised version.**

---

## Author Comment (AC3) · 20 Jun 2018

FINAL AUTHORS' STATEMENT IN RESPONSE TO REVIEWERS' COMMENTS We would like to thank both reviewers for their comments and suggestions on our manuscript. We have gone through the comments and suggestions and have detailed below how we intend to modify the manuscript to address these comments. We are confident that these changes can be achieved and will lead to a greatly improved, revised manuscript. Stephen Chastain, Dr. Karen Kohfeld, Dr. Marlow Pellatt

REVIEWER #1 POINT-BY-POINT RESPONSES #1. REVIEWER 1: "...I have a major problem with the estimate of the mass accumulation rate. There is no details on 210Pb data and not enough on the dating method.... "

[Figure]

"...the authors use an another technic, the alpha spectrometry. The problem of this method is that it measures 210Pb only. But the 210Pb-dating method is based on the decay of the excess Pb, ie the fraction of 210Pb not supported by its radioactive parent (226Ra) in sediment. This implies the authors made assumptions to estimate this supported fraction, this information is not given in the article. How the authors determine this supported value ? did they use the same value for the 4 cores. What is the error associated witih the assumption ? In addition, 210Pb/210Pbxs are not presented which is a critical aspect as these data define the SAR. The method used to estimate SAR/MAR is also not enough detailed."

RESPONSE: We will revise tables C2-C6 so that they include the unsupported 210Pb activity (and standard deviations), %C, and SCD for each subsample. We note that our analysis follows published methods using 210Po alpha counting, assuming that 210Po and 210Pb are in secular equilibrium (e.g. Binford 1990). This method also assumes that no unsupported 210Pb is present at the depth of lowest observed activity. (Additional measurements of 226Ra were outside the budget of the project.) The minimum 210Po activity at the base of each core was used to estimate background and was not the same for all cores. This method has been used with alpha counting to date sedimentary cores in previous studies (e.g., Brossier et al., 2014; Chambers et al., 2017; Galka et al., 2017; Greiner et al., 2013, Kolker et al. 2009; Wachnicka et al., 2013). We will clarify these points in the methods section.

**2. REVIEWER 1: "...From the CSR (constant rate of supply) model) based on the inventories, it is possible to calculate directly age of each layer, and then to estimate SAR and MAR, such values would have been interesting to discuss also (temporal trends, potential change in accretion regarding sea level rise)...."**

RESPONSE: As described also for Reviewer 2 (point #3), We will add a figure showing the changes in SAR and CAR with time and describe this figure in Results and Discussion sections. Tables C2-C6 will be modified to include SAR and CAR estimates for each core subsample.

**3. REVIEWER 1: "... I do not understand why the authors speech two times about 137Cs, there is useless. In fact in such environnements, where accretions could have change, 210Pb IN EXCESS is indeed appropriate, 137Cs is interesting only to validate the chronology.."**

RESPONSE: A similar comment was made by Reviewer 2. Our intent was to point out that 210Pb has not been used extensively for dating marsh sediments on the Pacific Coast, and there is evidence that 137Cs produces slightly (but not significantly) higher estimates of CAR (ie. Callaway et al. 2012). Given that dating is only one source of uncertainty in this analysis, we will shorten the section comparing 137Cs and 210Pb and instead incorporate this point into a new discussion of sources of uncertainty in our estimates (See response to reviewer 2, #4).

**4. REVIEWER 1: "The second problem is the sampling. ... the authors determine SAR only on 4 cores sampled in different systems. Do they assume there is no change in sediment according to the position along the transects ? What about the morphology along these transect ? Regarding the purpose of the article, I would have expect to have a higher number of cores on which 210Pb was determined in order to obtain more reliable SAR and then CAR. Whereas 210Pb is already mentioned in the abstract, there is no data of this radionuclides nor figures presenting profiles with depth. Considering the objectives of the article, that imply to know rather precisely SAR/MAR in order to calculate CAR, the number of dated cores is also too weak to be representative of the different systems."**

RESPONSE: We acknowledge that our sample size of dated cores was small, but we were financially constrained from submitting more cores for dating. In our revised manuscript, we will include a figure showing core stratigraphies of all cores sampled in the Clayoquot Sound marshes. This figure will demonstrate that most core stratigraphies were both relatively simple and, more importantly, very similar between sites. This will strengthen our argument that our choice of cores for 210Pb dating sampled representative areas that can be compared across marshes, as almost all cores progressed through the same sequences of topsoil, peat, and sand without major varia-tion. This justification will be described in the methods section; uncertainties associ-ated with this assumption will be described in the Discussion.

REVIEWER #2 POINT-BY-POINT RESPONSES

**1.REVIEWER 2: "I have major concerns about how carbon accumulation rates (CAR) are estimated. First, authors only estimate CAR in a total of five cores collected at 4 marshes, although they sampled a total of 34 sediment cores for C stock determination. The authors do not explain why only these cores were dated, or whether other cores were also analyzed by 210Pb but could not be dated. Mixing, erosion or changes in sedimentation are common processes in coastal sediments, and could lead to the alteration of sediment records, hence 210Pb concentration profiles (Ruiz-Fernández and Hillaire-Marcel, 2009). However, altered 210Pb profiles, although not datable, are results themselves."**

RESPONSE: In our revised text we will clarify that we selected cores for 210Pb dat-ing based their representative stratigraphies (ie. the depth of refusal was within a sand layer with %C values near zero, showing that the core sampled the full range of carbon accumulation). As stated in response to Reviewer 1 (point #4), we intend to clarify that these cores are representative of the general stratigraphic conditions of each marsh by including a figure that compares all core stratigraphies. We will justify our choice of cores for 210Pb dating based on this stratigraphic representativeness in the methods section; this section will also reference that we include all 210Pb profiles in the appendix, and that none demonstrate any noticeable mixing, erosion or other, post-depositional changes. We will also describe uncertainties associated with this as-sumption in the Discussion section. This uncertainty includes the fact that 3 cores out of 34 appear not to have reached the minimum %C depth.

**2. REVIEW 2: "The authors do not report data on total or excess 210Pb specific ac-tivities and no explanations are given regarding the determination of supported 210Pb,**

which might vary between marshes but also along the depth of their sediment profiles, especially if soils consist of three marked layers, topsoil, peat, and sand/clay (section 3.1, line16-17)."

AND

"The application of the CRS dating model to estimate SAR is unclear and some arguments should be provided regarding the election and application of this model. To apply this model certain assumptions must be met, for instance, this model is based in excess 210Pb inventories, which implies that the excess 210Pb horizon should have been reached in all dated cores. Without 210Pb data, this is impossible to evaluate. In addition, the CRS model provides estimates of SAR at each sediment layer rather than average sedimentation rates for the last century. In the main text Chastain et al. report average SARs at each core but do not explain how this average is estimated or if they have normalized SARs to a certain age-depth."

RESPONSE: As described in our response to Reviewer 1 (point #1), we will include all relevant 210Pb activity data in Tables C2-C6. By including the 210Po/210Pb activity in these tables, these concerns can be addressed.

**3. REVIEWER 2: "A results section showing 210Pb concentration profiles, 210Pb inventories and estimated fluxes should be included in the paper, this is important to evaluate whether the dating model applied is valid and to discuss the uncertainties associated to the estimation of ages and SAR. This will be included in the revised manuscript."**

RESPONSE: Figures showing 210Pb concentration profiles, 210Pb inventories, and estimate SAR and CAR fluxes (the latter described in point #2) will be provided with the manuscript and described in the Results and Discussion. We will include the 210Pb data in the revised appendix tables C2-C6. We will also modify our discussion section to include a discussion of the uncertainties of age estimations, as well as other sources of uncertainty. This will be in addition to the additional section discussing changes in

sedimentation and CAR over time.

**4. REVIEWER 2: "In the current version of the manuscript the authors include a section comparing 210Pb and 137Cs dating, which I believe is unnecessary; the authors did not analyze 137Cs in their cores and 137Cs is most commonly used to validate 210Pb chronologies. There are many aspects that can bias SAR and CAR high, for instance the presence of sediment mixing in 210Pb concentration profiles. My recommendation to the authors is to look critically at their 210Pb data and discuss the uncertainties related to their age-depth models, SAR and CAR estimates."**

RESPONSE: As described in response to Reviewer 1 (point #3), we will shorten the section comparing 137Cs and 210Pb, and include it in the new discussion section describing of sources of uncertainty.

**5. REVIEWER 2: "Second, to estimate CAR authors use sediment accumulation rates (SAR) which they multiply by the soil carbon density (SCD). While they acknowledge that sediment compaction occurred during coring and so they correct SAR for potential compaction, they do not correct SCD for such. This might lead to an overestimation of CAR. The authors estimate SCD multiplying the percent carbon content (%C) by the soil dry bulk density (DBD). While the rationale behind this is correct, soil DBD should be corrected for core compaction prior to the estimation of SCD. The mass contained in one cc volume of soil after coring occupies a greater volume in the field (before compaction occurs). Related to core compaction, I disagree with the statement in equation 8 used to estimate the uncompacted depth of a given subsample. Let's assume the recovered core length is 50 cm and the core penetration is 100 cm. This would result in a correction factor of 0.5 following equation 7. Then, if the correction factor is applied in equation 8 and is multiplied by the subsample depth (i.e., 1 cm-thick slice) this do not result in an uncompacted depth, please revise. In addition, compaction would unlikely have been linear throughout the soil column due to the presence of different soil layers (topsoil, peat, and sand/clay), which may show different degrees of compaction. For this reason, any variable that is sensitive to soil compaction**

such as DBD, SCD and SAR should not be used for the determination of CAR or C stocks. Variables such as the mass depth (m) (or mass per unit of area; g cm-2) and mass accumulation rate (MAR; g cm-2 yr-1) are not affected by soil compaction, then should be used instead of DBD and SAR to avoid the propagation of errors in the determination of CAR or C stocks (see below). My recommendation to the authors is to recalculate CAR as:

CAR (g C m-2 yr-1) = MAR (g cm-2 yr-1) x %C

Where the % C is not the average percentage of C along the sediment column but the fraction of the accumulated mass of C (gC cm-2/ g soil cm-2), estimated from the sum of the sediment layers accumulated over a period t = 100 yr, which should be approximately where the excess 210Pb horizon is reached."

RESPONSE: We will recalculate the accumulation rates using this approach, and update the Figures, Methods , Results, and Discussion accordingly.

**6. REVIEWER 2: "To finish with concerns about CAR estimates, I think differentiation between low and high marsh CAR is not possible with only one CAR estimate for a low marsh. The authors indeed acknowledge this at the end of the manuscript in section 4.4, line 3-4. I believe this should be said upfront. Accordingly, comparisons of Clayoquot Sound CAR with other salt marshes should be based only on high marsh CAR estimates reported at the other study sites. Final recommendation to the authors would be to avoid estimating total CAR for a marsh with only a dated core as the high marsh core CAR times the total marsh area (this is represented as a crosshatched column in figure 4). The latter is probably unlikely according to the results presented: lower C stocks in low marsh cores and low CAR in the single low marsh core."**

RESPONSE: We will remove this comparison between low and high marsh. Additionally, we will also modify Figure 4 to remove the "high marsh CAR * total marsh area" column.

**BGD**

**7. REVIEWER 2: Specific comments on the estimation of C stocks: Similarly, estimation of C stocks should be done using soil mass depth (g cm-2) rather than SCD multiplied by the thickness of soil slices, i.e., 1 cm, which is affected by soil compaction. My recommendation to the authors is to recalculate C stocks using the soil mass per unit area (m) rather than the sum of all sections DBD x %C x 1 cm. The soil mass per unit area at each layer is not affected by compaction or by inaccurate slicing. It is estimated by dividing the dry sample mass by the area sampled by the core tube, which is the cross-sectional area of its inner diameter (D), _(D/2)ËĘ2: Cstock_core=P (DW=(_rËĘ2 )) x %C"**

RESPONSE: We will re-calculate the stocks using this method and update the methods, results, and figures accordingly.

**8. REVIEWER 2: "The second problem here is the computation of overall averages when the averaged values are computed over a number of estimates that are different at each site or when the area each marsh represents is not the same. - The mean C stock at a marsh (Cstockmarsh) should be calculated as the weighted average of the mean Cstockscore estimated in the low marsh area and the mean of those estimated in the high marsh area, being the weights, the area made by low and high marsh at each individual marsh. - Then, the average C stock of low marshes at Clayoquot Sound (Cstock-LowCS) also should be a weighted average, with weights being the low marsh area of each individual marsh. Same for CstockHighCS."**

RESPONSE: In our original manuscript, we calculated BOTH the per-hectare marsh carbon stock, as well as multiplied the weighted high/low marsh averages by the estimated surface area of the marshes to calculate total marsh carbon stock (i.e., the total marsh carbon stock in Figure 4b). We will clarify in the text that these total marsh carbon stocks represent weighted averages.

**9. REVIEWER 2: "The authors use the depth of refusal (DoR) as a measure of the maximum depth of organic accumulation. C stocks are then estimated down to this**

depth (average 27.6 cm) and compared with those of global estimates (which some are estimated down to 1 m and others extrapolated to the same depth, 1 m). The authors conclude the C stocks at Clayoquot Sound are lower than those globally, but this is not a fair comparison. DoR is relative to the equipment being used and to the type of soils, therefore I feel that any comparisons made without standardizing all sites to a certain depth/mass depth (preferably) can be misleading. Rather than extrapolating their measurements to 1 m, the authors could normalize global estimates to 30 cm or perhaps to a certain mass depth, which would be the most consistent to establish comparisons (see Wendt and Hauser, 2013). As well, authors could discuss differences on C stocks based on %C, DBD and CAR rates found globally and at other regions such as eastern Canada, the Pacific Coast of the United States and Mexico. "

RESPONSE: We will clarify that our depth of refusal did not reflect a limitation of the equipment. The DoR was used as a proxy for the depth at which organic carbon reached 0 %, and in most cores this depth corresponded with a change in substrate from peat and peaty sand to dense sand, gravel, or clay. We will clarify this in our discussion section on global comparisons, and note that our average depth of refusal at 26.7 cm is close to a 30 cm normalized stock estimate depth.

**10. REVIEWER 2: "Authors should take action on the points listed above and revisit their calculations to provide more consistent estimates of C stocks and CAR. As well, they should discuss their results, perhaps, with more emphasis on C stocks and intra marsh variability (for which they have a good dataset), while presenting CAR results in a more local scale, avoiding upscaling to the Pacific coast of Canada. Instead, I encourage the authors to discuss temporal trends in C accumulation at the dated marshes if 210Pb profiles allow so. "**

RESPONSE: We plan to add analysis of local, intra-marsh variability to our discussion section. We will expand on the comparison of soil characteristics between our sites and to remove our comparison of low and high marsh zones. Furthermore, we will modify the boxplots of existing figure 5 to compare these six soil characteristics (depth

of refusal, DBD, estimated carbon stock per hectare, marsh area, percent carbon, and soil carbon density) between each site.

CONCLUSION: We are grateful for the helpful comments from both anonymous reviewers and are confident that we can address their concerns with further revisions to the data tables, figures and text (Methods, Results, and Discussion). Both reviewers' concerns with 210Pb data, dating uncertainties, and model selection can be addressed by including this information in appendix tables C2-C6 of dating results. Figures 2 and 3 will be modified to indicate which of the cores were selected for dating. Section 3.4 and Figures 4 and 5 will be amended to include a comparison of soil characteristics between the seven marshes rather than between high and low marsh. Figure 4 will also be altered to remove the column of high marsh CAR * total marsh area in chart (d). Figure 6 will be modified as needed with the recalculated CAR results, and with an additional caption item differentiating our 210Pb dates from other studies using 137Cs. The Methods section will be changed to describe the new methodology used for carbon stock and accumulation rate calculations as per reviewer #2's suggestions. Equations 4-8 will be changed to reflect this change. Section 2.5 on carbon accumulation rates will also include a short description of the representativeness of cores selected for dating. The Results in Sections 3.2 and 3.3 will be updated with the result of the new method of calculating stocks and accumulation rates as per reviewer #2's suggestions. The comparison of high and low marsh will be reduced to a single sentence. Discussion sections will be modified as follows: First, using Reviewer #2's recommended approach to estimating CAR and C stocks (independent of dry bulk density), we will compare carbon stocks with external studies using this method, which will be added into section 4.2. We will include a short discussion of intra-marsh variability in soil characteristics, particularly carbon content. Second, following both reviewers' suggestions, we will add a new discussion section on the changes in sediment accumulation and CAR over time from all five cores. Third, we will add to section 4.4 to expand upon our discussion of all sources of uncertainty in our estimates, including a mention that the sample size of dated cores was small (although we will also mention the lack of

visible alterations to 210Pb profiles). Among non-dated cores, we will mention that 3 of the 34 cores did not reach minimal %C values, which would result in a slight under-estimate of %C. However, all dated cores did reach minimal %C ($\sim$0 %), which should strengthen our argument that we sampled representative areas for 210Pb dating, and that our CAR estimates are therefore applicable to all sites. Finally, the section comparing 210Pb and 137Cs dating will be removed and replaced with a single sentence in the section 4.4 discussion of uncertainty.

References

Binford, M.W.: Calculation and uncertainty analysis of 210Pb dates for PIRLA project lake sediment cores, Journal of Paleolimnology, 3, 253-267, 1990. Brossier, B. F. Oris, W. Finsinger, H. Asselin, Y. Bergeron, A. A. Ali, Using tree-ring records to calibrate peak detection in fire reconstructions based on sedimentary charcoal records, The Holocene 24(6): 635-645, 2014. Chambers, F. A. Crowle, J. Daniell, D. Mauquoy, J. McCarroll, N. Sanderson, T. Thom, Ph. Toms, J. Webb, Ascertaining the nature and timing of mire degradation; using palaeoecology to assest future conservation management in Northern England, AIMS Environmental Science, 4(1): 54-82, 2017. Gałka, M, Szal, M, Watson, EJ et al. (6 more authors). Vegetation Succession, Carbon Accumulation and Hydrological Change in Subarctic Peatlands, Abisko, Northern Sweden. Permafrost and Periglacial Processes, 28 (4). pp. 589-604. ISSN 1045-6740, 2017. Greiner, J.T., McGlathery, K.J., Gunnell, J., & McKee, B.A.: Seagrass restoration enhances "blue carbon" sequestration in coastal waters, PloS one, 8, (8), 2013. Kolker, A. S. et al. High-resolution records of the response of coastal wetland systems to long-term and short-term sea-level variability, Estuarine, Coastal, and Shelf Science, 84(4): 493-508, 2009. Wachnicka, A., L. S. Collins, E. E. Gaiser, Response of diatom assemblages to 130 years of environmental change in Florida Bay (USA) Journal of Paleolimnology 49 (1): 83-101, 2013.